# Few-Shot Closed-Loop Neural System Identification via Meta-Learning

## Abstract

We study few-shot closed-loop neural system identification, where the goal is to recover an open-loop dynamics model from limited feedback-controlled data. In this setting, the observed inputs are generated through output feedback, so the collected trajectories are shaped by both the plant dynamics and the controller. This feedback-dependent data generation can bias direct prediction-error learning of the plant dynamics, and the difficulty is further amplified when only scarce target data are available for adaptation. To address this problem under continuous target trajectories, we propose Meta-ICI, which meta-learns a transferable initialization for an ICI-compatible operator and reconstructs the open-loop plant model through the known controller. We further extend Meta-ICI to fragmented target adaptation, where only scattered one-step transitions are available instead of continuous trajectories. This extension yields Fast Meta-ICI for fully observable systems, using fragmented transitions to support accurate long-horizon rollouts. To instantiate Fast Meta-ICI, we design a Schur-Koopman model that enforces the latent spectral-radius constraint during unconstrained optimization. Experiments on partially observable Liénard systems and fully observable nonlinear pendulum systems show that Meta-ICI improves few-shot adaptation and Fast Meta-ICI enables non-divergent long-horizon rollouts from fragmented target data.

## 1 Introduction

Dynamics models serve as a core component of modern machine learning systems for sequential decision making, including model-based reinforcement learning (Chua et al., 2018), model predictive control (Zhou et al., 2025), robotics (Kaufmann et al., 2023), and sequential time-series modeling (Chen et al., 2018). To successfully drive these decision-making processes, models cannot merely fit passively logged trajectories; they must actively support rollouts, planning, and counterfactual predictions. Specifically, the model should predict how the system would evolve under input sequences or excitation distributions different from those observed during data collection (Chua et al., 2018; Zhou et al., 2025). Neural-network-based dynamics models provide data-driven and differentiable parameterizations for this purpose, making them highly useful when first-principles models are unavailable, incomplete, or insufficiently accurate (Pillonetto et al., 2025).

In practice, obtaining the ideal data for learning such a model is often difficult. Many physical systems cannot be freely excited in open loop because they must remain stabilized by an existing feedback policy, or controller (Forssell & Ljung, 1999). Even under feedback control, informative identification experiments can be costly, as excitation signals consume operating time and may perturb the system away from normal operation (Ljung, 1998). Consequently, the available data for a target system may be limited, noisy, short, or fragmented rather than consisting of long continuous trajectories. This leads to the practical problem of few-shot closed-loop system identification, where the goal is to recover the open-loop dynamics of the plant from a small amount of feedback-controlled data.

Few-shot closed-loop system identification is more difficult than standard few-shot dynamics learning. Standard squared prediction-error training implicitly assumes that the prediction noise is orthogonal to the variables used by the predictor. In closed-loop settings, this condition is generally violated because the input sequence is generated through feedback from noisy observations, making the input statistically coupled with

disturbances and measurement noise (Dinkla et al., 2026). As a result, directly minimizing prediction error on closed-loop trajectories can yield a biased estimate of the plant dynamics. The few-shot setting further amplifies this issue, since only a small amount of target data is available to adapt a neural network and distinguish plant-specific dynamics from feedback-induced correlations.

Closed-loop identification methods explicitly account for the feedback mechanism instead of treating feedback-controlled trajectories as ordinary supervised data (Srivastava et al., 2023; Zhu et al., 2021; Boroujeni et al., 2025). In particular, Internal Controller Identification (ICI) reformulates closed-loop system identification as the learning of a stable internal operator associated with the known controller (Boroujeni et al., 2025). This provides a way to address feedback-induced bias, but ICI is trained for each target system independently and requires sufficiently informative target-system data. Meta-learning provides a way to address this limitation by using related source systems to learn a transferable initialization (Chakrabarty et al., 2023; 2025; Clavera et al., 2019).

Motivated by the complementary roles of ICI and meta-learning, we propose Meta-ICI, a framework that meta-learns the initialization of an ICI-compatible internal operator across related closed-loop systems. The ICI structure accounts for the feedback mechanism and mitigates feedback-induced bias, while meta-learning enables the operator to be adapted from limited target data rather than trained independently from scratch. For a new target system, the adapted internal operator is converted into the corresponding open-loop dynamics through the ICI reparameterization. Because ICI is formulated in terms of input-output operators, Meta-ICI can also be instantiated with recurrent stable architectures for partially observable systems.

For fully observable systems, we further develop Fast Meta-ICI, which exploits the Markov property of the observed state. In this case, target adaptation can be performed using scattered one-step transitions rather than long contiguous trajectories, while the learned model still supports long-horizon counterfactual rollouts. This fragmented one-step adaptation without sequence unrolling is the sense in which we use the term Fast. Fast Meta-ICI addresses this mismatch by combining a Schur-Koopman (SK) architecture with a hybrid step-wise/sequence-wise objective based on MAML (Finn et al., 2017). The SK architecture provides a stable neural operator suitable for ICI, while the hybrid objective adapts from fragmented one-step data and meta-optimizes rollout accuracy.

Our contributions are summarized as follows.

1. We introduce few-shot closed-loop neural system identification as a meta-learning problem, where the goal is to recover open-loop dynamics from limited feedback-controlled data by leveraging related source systems. To the best of our knowledge, this is among the first attempts to study meta-learning for closed-loop system identification.

2. We propose Meta-ICI, a model-agnostic meta-learning framework for ICI-compatible stable neural operators. Meta-ICI transfers the initialization of the internal operator used by ICI, enabling adaptation from limited closed-loop data while preserving the closed-loop identification structure needed to address feedback-induced bias.

3. For fully observable systems, we design an ICI-compatible SK operator with a built-in spectral-radius constraint. The operator uses a Koopman-inspired latent linear representation and a Schur parameterization to preserve stability constraints during unconstrained gradient-based optimization, making it compatible with standard optimizers such as SGD and Adam. Moreover, we prove that, under the stated latent contraction and reconstruction-consistency assumptions, this architecture defines a finite-gain $\mathcal{L}_p$-stable operator.

4. We propose a hybrid meta-learning objective for Fast Meta-ICI that turns fragmented target data into rollout-oriented adaptation. Instead of requiring continuous target trajectories, the inner loop adapts from scattered one-step transitions. The outer loop then optimizes the post-adaptation model through multi-step rollouts, enabling the learned initialization to support long-horizon counterfactual prediction.

5. We empirically validate the proposed methods on partially observable Liénard systems and fully observable nonlinear pendulum systems. The results show that Meta-ICI substantially improves few-shot adaptation

over scratch training, and that Fast Meta-ICI produces non-divergent long-horizon rollouts under shifted input distributions, approaching the corresponding full-data reference performance.

## 2 Related Work

**Deep neural networks for system identification.** Deep neural networks have been widely studied as flexible model classes for nonlinear system identification. Pillonetto et al. (2025) provide a recent survey of deep networks for system identification, covering various architectures as well as the optimization and regularization issues that arise when fitting dynamic models from input-output data. Beintema et al. (2023) proposed SUBNET for continuous-time nonlinear state-space identification, using a neural encoder to estimate latent initial states from subsequences and short simulations to approximate the full simulation loss. Control-oriented applications have also adopted neural predictors as system identification modules. For example, Frison & Gölzhäuser (2024) use Transformer-LSTM architectures for building heating system identification and forecasting. For feedback-controlled data, Boroujeni et al. (2025) provide a neural closed-loop identification framework by reformulating the problem as the learning of a stable internal operator associated with the known controller. However, both conventional deep-network-based identification methods and ICI are typically trained for each target system independently and require sufficiently informative target-system data. In contrast, our work studies few-shot adaptation across related feedback-controlled systems by meta-learning the initialization of ICI-compatible operators.

**Meta-learning for system identification.** Meta-learning has recently been explored as a way to improve data efficiency in neural system identification by leveraging data from related systems. Chakrabarty et al. (2023) proposed one of the first gradient-based meta-learning frameworks for neural state-space identification, using MAML to learn an initialization from similar source systems that can be rapidly adapted to a new target system with limited data. This work focused on autonomous neural state-space models and demonstrated advantages over supervised learning and transfer learning under limited target data. Chakrabarty et al. (2025) extended this direction to physically constrained neural system identification, incorporating domain-specific constraints into neural state-space models and demonstrating few-shot adaptation on real-world case studies. Besides, Rufolo et al. (2025) introduced a distributionally robust meta-learning objective that prioritizes high-loss tasks, improving worst-case and out-of-distribution performance. Forgione et al. (2025) proposed manifold meta-learning, which learns a low-dimensional manifold in the parameter space of an over-parameterized neural network to reduce adaptation complexity while preserving expressive power. These works show that meta-learning is a promising tool for few-shot neural system identification. However, they are primarily developed for open-loop trajectory data, and do not address closed-loop identification. Our work extends this line by meta-learning ICI-compatible operators, enabling few-shot adaptation to recover open-loop dynamics from feedback-controlled data.

**Stable neural operators and stable dynamics models.** Stability constraints have been widely used to make learned dynamical models reliable for prediction and control. Revay et al. (2024) introduced Recurrent Equilibrium Networks (REN), a class of nonlinear recurrent models with built-in stability guarantees and contraction properties. REN are suitable for input-output system identification because they can represent history-dependent stable operators, and we use them as the stable operator class for the partially observable Meta-ICI setting. Kojima & Okamoto (2022) proposed learning deep input-output stable dynamics by enforcing a Hamilton-Jacobi inequality, directly targeting input-output stability of neural dynamical systems. Other stable neural sequence models, such as $\mathcal{L}_2$-stable structured state-space models (Massai et al., 2026), prescribe input-output $\mathcal{L}_2$ bounds, although such guarantees do not directly imply the $\mathcal{L}_p$-stability and contraction requirements used in the ICI framework. For stable Koopman-style latent dynamics, Fan et al. (2026) proposed a general unconstrained Schur-stable parameterization for Koopman embeddings, and Forootani et al. (2026) combined Transformer-based sequence modeling with Koopman-inspired latent linear dynamics to improve stability. In our work, stability is a requirement for compatibility with the ICI framework. While existing recurrent stable models (such as REN) offer stability guarantees and support unconstrained optimization, their inherent reliance on contiguous data sequences to infer hidden states makes them inapplicable when only fragmented data points are available for few-shot target adaptation. For the fully observable setting, we overcome this by designing a SK operator. It operates purely on independent dis-

crete state transitions, naturally accommodating fragmented adaptation data, while intrinsically maintaining the spectral-radius constraint under standard unconstrained gradient descent.

## 3 Problem Formulation

We consider a family of unknown discrete-time nonlinear plants observed under feedback control. For each task $\mathcal{T}_i$, the plant is represented as a strictly causal input-output operator $G_i$. For a finite horizon, the measured output is written as

$$y_t^{(i)} = G_{i,t}(u_{0:t-1}^{(i)}) + v_t^{(i)}, \qquad t = 0, \ldots, T,$$

where $u_t^{(i)} \in \mathbb{R}^{n_u}$ is the plant input, $y_t^{(i)} \in \mathbb{R}^{n_y}$ is the measured output, and $v_t^{(i)}$ denotes unmeasured disturbances and measurement noise. Equivalently, using sequence notation,

$$y^{(i)} = G_i(u^{(i)}) + v^{(i)}. \tag{1}$$

The plant operates in closed loop with a known causal output-feedback controller $K_i$ and an external reference signal $r_t^{(i)}$:

$$u^{(i)} = r^{(i)} + K_i(y^{(i)}). \tag{2}$$

Thus, the collected data consist of closed-loop trajectories

$$\mathcal{D}_i = \left\{ \left( r_{0:T-1}^{(i,j)}, u_{0:T-1}^{(i,j)}, y_{0:T}^{(i,j)} \right) \right\}_{j=1}^{N_i}.$$

Here, $N_i$ denotes the number of trajectories for task $\mathcal{T}_i$. The index $i$ specifies the task, while $j$ indexes different trajectories within the same task, which may differ in initial condition, reference signal, excitation sequence or noise. We assume that the closed-loop interconnection is well posed and $\mathcal{L}_p$-stable, i.e., the mapping $(r^{(i)}, v^{(i)}) \mapsto (u^{(i)}, y^{(i)})$ is an $\mathcal{L}_p$-stable operator.

The objective is to identify an open-loop dynamics model $\hat{G}_\star$ for a new target plant $G_\star$ using only limited and possibly fragmented target closed-loop data. This differs from standard open-loop neural system identification methods (Chakrabarty et al., 2023; 2025; Rufolo et al., 2025), because the input $u_t$ is not chosen independently of the plant output. Since $u_t$ is generated through feedback from noisy measurements, the regressors used for prediction can be statistically coupled with disturbances and measurement noise. Consequently, directly fitting a neural model on closed-loop trajectories may produce biased models unless additional assumptions, excitation, or identification structure are used (Forssell & Ljung, 1999; Ljung, 1998; Boroujeni et al., 2025). In the few-shot setting, this difficulty is further compounded because the target data may provide only limited coverage of the state-input region needed for counterfactual open-loop rollouts.

We assume access to a collection of related source tasks $\mathcal{T}_i \overset{\text{i.i.d.}}{\sim} p(\mathcal{T}), i = 1, \ldots, M$, where each task corresponds to a closed-loop system with compatible input-output dimensions and known controller $K_i$. The tasks share a common structural class, such as related physical dynamics or common modeling assumptions, but may differ in their system parameters, nonlinearities, or operating conditions. Each source task provides a relatively richer closed-loop dataset $\mathcal{D}_i$. In contrast, the target task $\mathcal{T}_\star$ provides only a small adaptation dataset $\mathcal{D}_\star^{\text{adapt}}$, which may consist of a few short closed-loop trajectories or scattered one-step transitions rather than continuous trajectories.

Therefore, the few-shot closed-loop neural system identification problem is to use the source datasets $\{\mathcal{D}_i\}_{i=1}^M$ to learn a transferable model prior that can be rapidly adapted to the target system $\theta_\star' = \mathcal{A}_\alpha(\theta, \mathcal{D}_\star^{\text{adapt}})$, where $\theta$ denotes the shared initialization and $\mathcal{A}_\alpha$ denotes a small number of adaptation steps with learning rate $\alpha$. The adapted parameters $\theta_\star'$ are then used to construct an open-loop model $\hat{G}_\star$ for the target plant.

At the meta-training stage, each source dataset is split at the trajectory level into a support set and a query set, i.e., $\mathcal{D}_i = \mathcal{D}_i^{\text{sup}} \cup \mathcal{D}_i^{\text{qry}}, \mathcal{D}_i^{\text{sup}} \cap \mathcal{D}_i^{\text{qry}} = \emptyset$. The support set simulates the limited target data available during adaptation, while the query set evaluates the post-adaptation model. The loss used for adaptation need not be identical to the loss used for meta-evaluation. In particular, the support loss determines how the

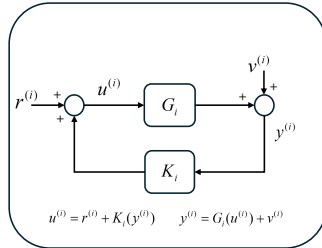 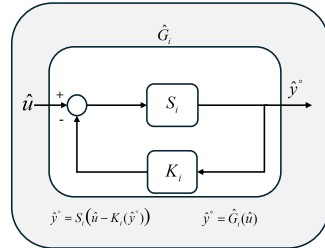

Figure 1: Illustration of the ICI framework. The left panel shows the closed-loop interconnection of the plant $G_i$ and the known controller $K_i$. The right panel shows the ICI representation, where the open-loop model $\hat{G}_i$ is induced by interconnecting the same controller $K_i$ with a stable operator $S_i$.

model is adapted from limited data, whereas the query loss determines which adapted models are preferred by the meta-objective.

Following the gradient-based meta-learning formulation of MAML (Finn et al., 2017), given an initialization $\theta$, the task-specific adapted parameter is obtained by a small number of gradient steps on a support loss. For one-step adaptation, this gives

$$\theta_i' = \mathcal{A}_\alpha(\theta, \mathcal{D}_i^{\text{sup}}) = \theta - \alpha \nabla_\theta \mathcal{L}_i^{\text{sup}}(\theta, \mathcal{D}_i^{\text{sup}}), \tag{3}$$

with the extension to multiple adaptation steps being straightforward. The meta-objective then evaluates the adapted parameter on a query loss:

$$\min_\theta \ \mathbb{E}_{\mathcal{T}_i \sim p(\mathcal{T})} \left[ \mathcal{L}_i^{\text{qry}} \left( \theta_i', \mathcal{D}_i^{\text{qry}} \right) \right]. \tag{4}$$

Different choices of $\mathcal{L}_i^{\text{sup}}$ and $\mathcal{L}_i^{\text{qry}}$ lead to different instantiations of the proposed framework. For general partially observable systems, adaptation must usually be performed on contiguous trajectories, since the model needs temporal context to infer unmeasured internal states. In this case, both the support and query losses are naturally sequence-based. For fully observable systems, however, each transition already contains the complete state information needed for one-step prediction. This makes it possible to use a step-wise support loss on scattered transitions during adaptation, while still using a sequence-wise query loss to select initializations that produce accurate long-horizon rollouts after adaptation.

## 4 Proposed Approach

This section presents the meta-learning framework for few-shot closed-loop neural system identification. We first introduce Meta-ICI, which combines the ICI reparameterization with gradient-based meta-learning. The ICI structure provides a closed-loop identification parameterization through a stable internal operator, while meta-learning provides an initialization that can be adapted from limited target data. We then introduce Fast Meta-ICI for the fully observable setting, where the availability of the full state allows target adaptation to use scattered one-step transitions instead of contiguous trajectories.

### 4.1 Meta-ICI Framework

For each task $\mathcal{T}_i$, the unknown plant $G_i$ operates in closed loop with a known causal controller $K_i$.

Our goal is to use a neural network to identify the open-loop dynamics of target system $G_\star$ from closed-loop data while retaining the prior knowledge that $K_\star$ stabilizes the true plant.

We build on the ICI parameterization, illustrated in Fig. 1. This figure is redrawn from (Boroujeni et al., 2025) to provide a clear understanding. The key idea is to represent the open-loop plant model through a stable internal operator and the known controller. Specifically, ICI introduces a operator $S_i \in \mathcal{L}_p^{SC}$ and defines the nominal model output $\hat{y}^\circ$ for a candidate open-loop input $\hat{u}$ by

$$\hat{y}^\circ = S_i \left( \hat{u} - K_i(\hat{y}^\circ) \right). \tag{5}$$

This interconnection induces the corresponding open-loop plant model $\hat{G}_i$ as the input-output map

$$\hat{G}_i(\hat{u}) := \hat{y}^{\circ}. \tag{6}$$

The usefulness of this construction follows the theorem (see Boroujeni et al., 2025, Theorem 1): If $K_i$ is incrementally $\mathcal{L}_p$-stable, then any $S_i \in \mathcal{L}_p^{SC}$ inserted into equation 5 defines a model $\hat{G}_i$ whose closed-loop interconnection with $K_i$ is $\mathcal{L}_p$-stable. Conversely, any strictly causal plant $G_i$ stabilized by $K_i$ admits such a representation for some $S_i \in \mathcal{L}_p^{SC}$. Thus, learning the plant within the class of systems stabilized by $K_i$ can be reduced to learning the internal operator $S_i$.

It remains to specify how $S_i$ is learned from closed-loop data. Following the indirect training strategy (Boroujeni et al., 2025), we exploit the closed-loop relation in equations 1, 2 and 5. For the closed-loop model, the noisy output can be written as

$$\hat{y} = S_i\big(\hat{u} - K_i(\hat{y} - v)\big) + v = S_i\big(r + K_i(\hat{y}) - K_i(\hat{y} - v)\big) + v.$$

When $S_i$ and $K_i$ are locally approximated by their linearizations, then $\hat{y} \approx S_i(r) + S_{i,L}(K_{i,L}(v)) + v$, where $S_{i,L}$ and $K_{i,L}$ denote the corresponding local linearizations. This linearized approximation holds well at high output signal-to-noise ratio (SNR) and high excitation SNR. This formulation separates $r$ and $v$ such that the identification of $S_i$ with closed-loop data becomes a standard open-loop identification, which motivates the indirect learning objective $r^{(i)} \mapsto y^{(i)}$ for estimating $S_i$.

We parameterize the internal operator by a stable neural operator $S_\theta$. For any task-specific dataset $\mathcal{D}$, e.g., $\mathcal{D}_i^{\sup}$, $\mathcal{D}_i^{\mathrm{qry}}$, or $\mathcal{D}_i$, a trajectory $\tau^{(i,j)} = \left(r_{0:T-1}^{(i,j)}, y_{0:T}^{(i,j)}\right)$ with its predicted output $\hat{y}_{0:T}^{(i,j)} = S_\theta\left(r_{0:T-1}^{(i,j)}\right)$, the sequence prediction loss for task $\mathcal{T}_i$ is defined as

$$\mathcal{L}_i^{\mathrm{seq}}(\theta, \mathcal{D}) = \frac{1}{|\mathcal{D}|} \sum_{\tau^{(i,j)} \in \mathcal{D}} \sum_{t=0}^{T} \left\| y_t^{(i,j)} - \hat{y}_t^{(i,j)} \right\|^2.$$

Meta-ICI applies the meta-learning formulation introduced in Section 3 to the ICI internal operator. For general partially observable systems, the model must use temporal context to infer hidden internal states. Therefore, both adaptation and meta-evaluation are performed on contiguous trajectories, which means that $\mathcal{L}_i^{\sup} = \mathcal{L}_i^{\mathrm{seq}}, \mathcal{L}_i^{\mathrm{qry}} = \mathcal{L}_i^{\mathrm{seq}}$.

After meta-training by equation 3 and equation 4, we get the learned initialization $\theta^\star$, which is adapted to a new target task using the limited target adaptation dataset:

$$\theta_\star' = \mathcal{A}_\alpha(\theta^\star, \mathcal{D}_\star^{\mathrm{adapt}}).$$

The final target open-loop model is then constructed through the ICI relation

$$\hat{G}_\star(\bar{u}) = \bar{y}, \qquad \bar{y} = S_{\theta_\star'}\big(\bar{u} - K_\star(\bar{y})\big), \tag{7}$$

where $\bar{u}$ denotes the input applied to the reconstructed open-loop plant.

**Remark 1.** The ICI representation is applied task-wise. Each task $\mathcal{T}_i$ may have its own known controller $K_i$, and the corresponding internal operator $S_i$ is defined with respect to this controller. Meta-ICI therefore does not require all tasks to share the same controller, nor does it meta-learn the controller itself. Instead, it meta-learns an initialization for the family of ICI-induced internal operators $\{S_i\}$. Successful transfer is assumed at the level of these induced operators, rather than at the level of the plant or controller alone. At target time, the adapted operator $S_{\theta_\star'}$ is combined with the target controller $K_\star$ through the ICI relation to construct the open-loop model.

This formulation is agnostic to the specific stable neural operator used to parameterize $S_\theta$. For partially observable systems, recurrent stable operators such as REN (Revay et al., 2024) are natural choices because they can represent history-dependent input-output maps. However, when the full state is observable, the Markov structure allows a more efficient instantiation in which adaptation is performed using scattered one-step transitions, while meta-training still evaluates long-horizon rollout behavior. The resulting fully observable instantiation is Fast Meta-ICI.

### 4.2 Fast Meta-ICI for Fully Observable Systems

We now specialize Meta-ICI to the fully observable setting. Throughout this subsection, we assume that the measured output is the full state $y_t^{(i)} = x_t^{(i)} \in \mathbb{R}^{n_x}$. In the general input-output setting, the internal operator $S_\theta$ is learned from reference-output trajectories and must use temporal history to represent hidden dynamics. When the full state is observable, this history dependence is no longer necessary. The ICI internal operator $S_i$ can instead be implemented as a Markovian one-step transition model

$$x_{t+1}^{(i)} = f_\theta \left( x_t^{(i)}, r_t^{(i)} \right),$$

where $f_\theta$ is the neural network implementation of $S_i$, $r_t^{(i)}$ is the same reference signal used as the input in Section 4.1.

#### 4.2.1 Schur-Koopman Operator

To instantiate the fully observable internal operator $f_\theta$, we need a neural transition model compatible with the $\mathcal{L}_p^{SC}$ requirement of ICI. A direct way to obtain stability is to enforce contraction of the map, for instance by applying spectral normalization to the neural network (Miyato et al., 2018). Although this strategy is general, it controls stability by requiring every one-step update to contract in a prescribed norm. This requirement can be stronger than necessary for stable dynamics with oscillations, transient amplification, or light damping, where the trajectory may exhibit local growth before eventually decaying (Trefethen & Embree, 2005).

For linear systems, stability is governed by the spectral radius of the state-transition matrix rather than by its spectral norm. A matrix with spectral radius smaller than one may still have spectral norm larger than one, allowing transient amplification while remaining asymptotically stable (Hespanha, 2018). The Koopman-inspired structure lets us exploit this distinction by lifting the nonlinear state into a latent space, evolving it with a linear operator, and decoding it back to the state space (Lusch et al., 2018).

The Koopman-inspired internal operator consists of an encoder $\phi_\theta : \mathbb{R}^{n_x} \to \mathbb{R}^{n_z}$, a decoder $\psi_\theta : \mathbb{R}^{n_z} \to \mathbb{R}^{n_x}$, a latent transition matrix $A_\theta \in \mathbb{R}^{n_z \times n_z}$, and a state-dependent input map $g_\theta : \mathbb{R}^{n_x} \to \mathbb{R}^{n_x \times n_u}$. The transition model is

$$x_{t+1} = f_\theta(x_t, r_t) = \psi_\theta(A_\theta \phi_\theta(x_t)) + g_\theta(x_t)r_t. \tag{8}$$

The spectral-radius constraint on $A_\theta$ is not sufficient to guarantee $\mathcal{L}_p$ stability of the nonlinear operator, because the encoder-decoder composition may distort the latent dynamics. The following theorem states the stability condition required by the system 8.

**Theorem 1.** *Consider the zero-initial-state system induced by equation 8. Assume that: (i) $\phi_\theta$ and $\psi_\theta$ are Lipschitz continuous with Lipschitz constant $L_\psi$ and $L_\phi$ respectively, and satisfy $\phi_\theta(0) = 0$ and $\psi_\theta(0) = 0$; (ii) $g_\theta$ is bounded on the compact operating domain, i.e., there exists $\Gamma > 0$ such that $\|g_\theta(x_t)\|_2 \leq \Gamma$ for all admissible $x_t$; and (iii) the latent autonomous map*

$$F_\theta(z_t) = \phi_\theta(\psi_\theta(A_\theta z_t)) \tag{9}$$

*is a strict contraction in some norm $\|\cdot\|_P$, namely there exists $\lambda \in (0,1)$ such that $\|F_\theta(z_t)\|_P \leq \lambda\|z_t\|_P, \forall z_t$. Then the operator mapping the input sequence $r \in \ell_p$ to the state sequence $x \in \ell_p$ is $\mathcal{L}_p$-stable for all $p \in [1, \infty]$. That is, there exists a constant $C > 0$ such that $\|x\|_{\ell_p} \leq C\|r\|_{\ell_p}$.*

The proof is provided in Appendix C.

Theorem 1 shows that the key remaining requirement is the contraction of the nonlinear latent autonomous map 9. We now relate this condition to two implementable design choices.

Let $z_{t+1}^A = A_\theta z_t$ and $\Delta_\theta(z_{t+1}^A) = \phi_\theta(\psi_\theta(z_{t+1}^A)) - z_{t+1}^A$. Then $F_\theta(z_t) = A_\theta z_t + \Delta_\theta(A_\theta z_t)$.

Suppose that the latent transition is Schur stable, i.e., $\rho(A_\theta) < 1$. By the spectral-radius lemma (Horn & Johnson, 2012), there exists an induced norm $\|\cdot\|_P$ and a constant $\rho_P \in (0,1)$ such that $\|A_\theta z_t\|_P \leq \rho_P\|z_t\|_P$.

In addition, if the encoder-decoder residual $\Delta_\theta(\cdot)$ is sufficiently small, i.e., $\|\Delta_\theta(A_\theta z_t)\|_P \leq \delta\|A_\theta z_t\|_P, \delta \geq 0$, then applying the triangle inequality gives

$$\|F_\theta(z_t)\|_P \leq \|A_\theta z_t\|_P + \|\Delta_\theta(A_\theta z_t)\|_P \leq (1+\delta)\|A_\theta z_t\|_P \leq (1+\delta)\rho_P\|z_t\|_P.$$

Whenever $\delta$ is small enough such that $\lambda = (1+\delta)\rho_P < 1$, the nonlinear latent autonomous map 9 satisfies the contraction condition required in Theorem 1.

We add

$$\mathcal{L}_{\text{rec}}(\theta, \mathcal{D}) = \frac{1}{|\mathcal{Z}_\mathcal{D}^A|} \sum_{\tilde{z} \in \mathcal{Z}_\mathcal{D}^A} \|\tilde{z} - \phi_\theta(\psi_\theta(\tilde{z}))\|_2^2, \qquad \mathcal{Z}_\mathcal{D}^A = \{A_\theta\phi_\theta(x_t) : (x_t, \cdot, \cdot) \in \mathcal{D}\}. \tag{10}$$

Since the latent space is finite-dimensional, norm equivalence implies that controlling the Euclidean reconstruction residual also controls the same residual in $\|\cdot\|_P$ up to constant factors (Horn & Johnson, 2012). Thus, the reconstruction loss serves as a soft penalty on the encoder-decoder residual involved in the sufficient contraction condition.

It remains to enforce the spectral-radius condition $\rho(A_\theta) < 1$. To this end, we parameterize $A_\theta$ through a real Schur form

$$A_\theta = Q_\theta \Xi_\theta Q_\theta^\top, \qquad \Xi_\theta = D_\theta + U_\theta.$$

Here, $Q_\theta$ is an orthogonal matrix and $\Xi_\theta$ is a real quasi-upper triangular matrix. The block diagonal part $D_\theta$ contains $2 \times 2$ rotation-scaling blocks

$$R_k = \rho_k \begin{bmatrix} \cos\omega_k & -\sin\omega_k \\ \sin\omega_k & \cos\omega_k \end{bmatrix}, \qquad \rho_k = \bar{\rho}\,\sigma(a_k), \qquad 0 < \bar{\rho} < 1, \tag{11}$$

where $k = 1, \ldots, \lfloor n_z/2 \rfloor$ indexes the $2 \times 2$ real Schur blocks, $\omega_k \in \mathbb{R}$ is a learnable rotation angle, $a_k \in \mathbb{R}$ is an unconstrained learnable scalar controlling the block radius, and $\sigma$ is the sigmoid function. If $n_z$ is odd, the last $1 \times 1$ block is parameterized as $\lambda_{\text{last}} = \bar{\rho}\tanh(a_{\text{last}})$, with $a_{\text{last}} \in \mathbb{R}$, so that $|\lambda_{\text{last}}| < \bar{\rho}$. The matrix $U_\theta$ contains learnable entries only in the strictly block upper-triangular part associated with this Schur block partition. Consequently, $\Xi_\theta$ remains block upper triangular, and its eigenvalues are exactly the eigenvalues of the diagonal blocks in $D_\theta$. Since each block has eigenvalue magnitude smaller than $\bar{\rho}$, the construction guarantees $\rho(A_\theta) = \rho(\Xi_\theta) < \bar{\rho} < 1$.

This parameterization converts the constrained search over latent transition matrices into an unconstrained optimization over Schur parameters. It allows the SK operator to be trained with standard neural-network optimizers such as SGD and Adam.

**Remark 2.** The parameterization in equation 11 can be viewed as a structured real-Schur parameterization. For simplicity, when $n_z$ is even, we use only $2 \times 2$ rotation-scaling blocks for all diagonal blocks. Each block has eigenvalues $\rho_k e^{\pm i\omega_k}$, so the spectral-radius constraint is enforced simply by choosing $\bar{\rho} < 1$. This design is less general than a complete Schur-stable matrix parameterization, such as the direct parameterization of (Fan et al., 2026), because real eigenvalues are represented as paired modes when only $2 \times 2$ blocks are used. In our Koopman-style latent model, this restriction is less severe because the latent dimension is typically overcomplete, and redundant latent coordinates could be suppressed by the encoder-decoder maps. Therefore, the main purpose of this structured form is to provide a lightweight, unconstrained, and oscillation-aware stable latent transition for the ICI-compatible fully observable setting.

### 4.2.2 Hybrid Step-wise/Sequence-wise Meta-Objective

The fully observable setting allows the support and query losses to use different data formats. Therefore, the support set may consist of scattered one-step transitions, $\mathcal{D}_i^{\text{sup}} = \left\{ (x_t^{(i,j)}, r_t^{(i,j)}, x_{t+1}^{(i,j)}) \right\}$, while the query set consists of contiguous rollout segments, $\mathcal{D}_i^{\text{qry}} = \left\{ (x_0^{(i,j)}, r_{0:T-1}^{(i,j)}, x_{1:T}^{(i,j)}) \right\}$.

Consequently, the support loss is a one-step-wise transition loss:

$$\mathcal{L}_i^{\text{step}}(\theta, \mathcal{D}_i^{\text{sup}}) = \frac{1}{|\mathcal{D}_i^{\text{sup}}|} \sum_{(x_t, r_t, x_{t+1}) \in \mathcal{D}_i^{\text{sup}}} \|x_{t+1} - f_\theta(x_t, r_t)\|^2 + \lambda_{\text{rec}}\mathcal{L}_{\text{rec}}(\theta, \mathcal{D}_i^{\text{sup}}). \tag{12}$$

The adapted parameter, $\theta_i' = \mathcal{A}_\alpha^{\mathrm{step}}(\theta, \mathcal{D}_i^{\mathrm{sup}})$, for task $\mathcal{T}_i$ is obtained by applying one or more gradient steps on this support loss.

The query loss evaluates the adapted model through multi-step rollout. That is, given an initial state $x_0$ and a reference sequence $r_{0:T-1}$, the predicted rollout is generated by

$$\hat{x}_0 = x_0, \qquad \hat{x}_{t+1} = f_{\theta_i'}(\hat{x}_t, r_t), \qquad t = 0, \dots, T-1.$$

The sequence-wise query loss is

$$\mathcal{L}_i^{\mathrm{roll}}(\theta_i', \mathcal{D}_i^{\mathrm{qry}}) = \frac{1}{|\mathcal{D}_i^{\mathrm{qry}}|} \sum_{(x_0, r_{0:T-1}, x_{1:T}) \in \mathcal{D}_i^{\mathrm{qry}}} \sum_{t=0}^{T-1} \|x_{t+1} - \hat{x}_{t+1}\|^2 + \lambda_{\mathrm{rec}} \mathcal{L}_{\mathrm{rec}}(\theta_i', \mathcal{D}_i^{\mathrm{qry}}). \tag{13}$$

Fast Meta-ICI uses the step-wise loss for adaptation and the rollout loss for meta-evaluation, i.e., $\mathcal{L}_i^{\mathrm{sup}} = \mathcal{L}_i^{\mathrm{step}}, \mathcal{L}_i^{\mathrm{qry}} = \mathcal{L}_i^{\mathrm{roll}}$.

A step-by-step pseudocode description of the hybrid-loss computation is provided in Appendix B.2.6.

After meta-training, the learned initialization $\theta^\star$ is adapted to the target system using the fragmented target adaptation set $\mathcal{D}_\star^{\mathrm{adapt}} = \{(x_t, r_t, x_{t+1})\}$, yielding $\theta_\star' = \mathcal{A}_\alpha^{\mathrm{step}}\left(\theta^\star, \mathcal{D}_\star^{\mathrm{adapt}}\right)$. The adapted model is then evaluated or deployed through the ICI open-loop reconstruction in equation 7.

## 5 Experiments

We design our empirical evaluation to validate the properties of the proposed frameworks. Specifically, our experiments aim to answer the following questions:

- **Q1 (Data Efficiency in Partial Observability):** To what extent does the Meta-ICI framework using recurrent structures improve data efficiency when adapting to an in-distribution target task under partial observability?

- **Q2 (Fragmented Adaptation under Shifted Inputs):** Under fully observable conditions, can Fast Meta-ICI recover full-data-level long-horizon predictive accuracy from fragmented target data, even when evaluated on input families not seen during training?

Throughout the experiments, we compare the proposed few-shot adaptation methods against full-data references trained with abundant target-system data under the same model class and evaluation protocol. They serve as strong empirical baselines indicating the performance attainable by the same architecture when substantially richer target data are available.

Detailed experimental configurations are deferred to Appendices B.1 and B.2.

### 5.1 Efficiency under Partial Observability (Experiment 1)

To answer **Q1**, we evaluate the Meta-ICI framework on a family of partially observable Liénard systems. The nonlinear damping parameter defines the system family. We use $c = 1.57$ as the target task, which lies in the same parameter range $c \in [0.5, 2.0]$ used for the source tasks. Only the first state is measured, while the second state remains hidden. Detailed dynamics, controller, and signal-generation settings are provided in Appendix B.1.1.

For the stable operator $\boldsymbol{S}_\theta$, we employ a Contractive Recurrent Equilibrium Network (C-REN). As a recurrent architecture, it is naturally suited for partially observable settings by maintaining hidden states to infer unmeasured dynamics, while its parameterization intrinsically guarantees the strictly causal and stable condition $(\mathcal{L}_p^{SC})$ required by the Meta-ICI framework.

**Evaluation Protocol and Baselines.** We assess the predictive performance using the FIT score, defined as $\mathrm{FIT} = 100 \times \left(1 - \frac{\|\hat{y} - y\|_2}{\|y - \bar{y}\|_2}\right)$, where $y$ is the true trajectory, $\bar{y}$ is its mean, and $\hat{y}$ is the predicted trajectory.

A FIT of 100% indicates perfect prediction. For a rigorous comparison, all evaluated models are tested under identical initial conditions and excitation signals. All reported FIT values are the mean $\pm$ standard deviation computed over 5 random seeds. We compare our proposed **Meta-ICI** against two baselines: (1) **Full-Data ICI Reference**, trained on 100 target-system trajectories under the same C-REN architecture, and (2) **Scratch ICI**, a C-REN initialized with random weights and fine-tuned exclusively on the limited target data.

As summarized in Table 1, the results demonstrate that Meta-ICI consistently and significantly outperforms Scratch ICI across all regimes. Meta-ICI rapidly approaches the Full-Data ICI Reference (75.16%) even in extreme few-shot scenarios. With merely 3 trajectories, the performance reaches 64.13%. Furthermore, a larger data size significantly accelerates the initial performance gain: 10 trajectories achieve 34.73% compared to $-4.69\%$ for 1 trajectory at step 3, showing that richer data provides a better gradient direction. An extended adaptation sweep with more target trajectories and longer adaptation budgets is provided in Appendix B.1.5.

Table 1: Predictive FIT [%] results under partial observability. Meta-ICI achieves superior data efficiency. It exceeds 64% FIT with as few as 3 target trajectories, approaching the Full-Data ICI Reference trained on 100 target trajectories. Larger data sizes lead to faster early-stage improvement, and the consistent upward trend across adaptation steps highlights rapid convergence. In contrast, Scratch ICI remains ineffective under identical conditions. All reported values are the mean $\pm$ standard deviation across 5 random seeds.

| Method | Data Size | Adaptation Steps | | | | |
|---|---|---|---|---|---|---|
| | | **0** | **1** | **3** | **5** | **10** |
| **Full-Data ICI Reference** | 100 Trajs | | | $75.16 \pm 8.47$ | | |
| **Meta-ICI** (Ours) | 1 Traj | $-73.26 \pm 20.52$ | $-47.62 \pm 17.47$ | $-4.69 \pm 11.55$ | $26.95 \pm 7.56$ | $58.79 \pm 8.91$ |
| | 3 Trajs | | $-40.30 \pm 17.13$ | $14.12 \pm 9.98$ | $47.92 \pm 8.29$ | $64.13 \pm 12.07$ |
| | 5 Trajs | — | $-31.73 \pm 15.85$ | $33.47 \pm 7.94$ | $57.83 \pm 11.03$ | $64.15 \pm 12.63$ |
| | 8 Trajs | | $-34.04 \pm 16.31$ | $30.46 \pm 8.19$ | $57.48 \pm 10.61$ | $64.37 \pm 12.57$ |
| | 10 Trajs | | $-32.05 \pm 15.71$ | $34.73 \pm 7.84$ | $58.36 \pm 11.08$ | $64.30 \pm 12.38$ |
| **Scratch ICI** | 1 Traj | $-97.78 \pm 37.54$ | $-76.36 \pm 35.07$ | $-52.10 \pm 32.13$ | $-39.82 \pm 30.43$ | $-28.38 \pm 28.38$ |
| | 3 Trajs | | $-74.95 \pm 34.93$ | $-49.86 \pm 31.91$ | $-37.58 \pm 30.16$ | $-26.69 \pm 28.10$ |
| | 5 Trajs | — | $-73.51 \pm 34.76$ | $-47.30 \pm 31.56$ | $-34.81 \pm 29.69$ | $-24.41 \pm 27.51$ |
| | 8 Trajs | | $-74.17 \pm 34.83$ | $-47.94 \pm 31.63$ | $-35.12 \pm 29.73$ | $-24.25 \pm 27.45$ |
| | 10 Trajs | | $-73.21 \pm 34.69$ | $-46.55 \pm 31.37$ | $-33.91 \pm 29.42$ | $-23.69 \pm 27.16$ |

## 5.2 Robustness to Fragmented Data and Long-Horizon Stability (Experiment 2)

To answer **Q2**, we shift our focus to fully observable dynamics and evaluate the proposed Fast Meta-ICI method on a nonlinear pendulum benchmark. The central question is whether Fast Meta-ICI can identify an accurate long-horizon dynamics model from fragmented few-shot target data, while approaching full-data reference performance even under test inputs not seen during training. The target adaptation data consist only of scattered single-step transitions, whereas evaluation is performed through continuous 200-step open-loop rollouts. This deliberately separates the data format available for adaptation from the long-horizon predictive behavior required at test time.

**Protocol.** Training data are collected from closed-loop trajectories driven by randomized bounded multisine references, while testing uses open-loop zero-input responses and step responses. We refer to each fixed test condition as a test scenario. The test set contains ten test scenarios, including five zero-input scenarios with different initial angles and five step-response scenarios with different constant input amplitudes.

Fast Meta-ICI adapts from randomly sampled single-step target transitions. We use repeated runs to denote the five random seeds used for reporting mean and standard deviation. For Fast Meta-ICI, these seeds correspond to independently sampled fragmented target-adaptation sets. For the Full-Data SK-ICI Reference, they correspond to independent full-data training runs with different random initializations and minibatch orders. Each table entry is computed by first averaging over the five test scenarios within the same input family and then reporting the mean and standard deviation over the five repeated runs.

Detailed system parameters and data-generation settings are provided in Appendix B.2.1. The exact test scenarios, validation grids, and optimization hyperparameters are provided in Appendices B.2.2 and B.2.3.

**Compared models. Full-Data SK-ICI Reference** is a SK model trained with abundant continuous target-system trajectories under the same architecture and test protocol. **Fast Meta-ICI** uses the SK architecture with the Hybrid MAML initialization, then adapts on the fragmented target data. The comparison focuses on whether few-shot adaptation can approach this empirical full-data reference under qualitatively shifted test inputs. For completeness, we also ran a **Scratch SK** baseline with the same SK architecture and fragmented target adaptation data but random initialization. This baseline uses no meta-learned initialization and no learned inner-loop learning rates.

**Metric and model selection.** Because zero-input and small-amplitude responses can have low signal energy, FIT scores may become unstable under this protocol. We therefore report NRMSE separately for $\theta$ and $\omega$. Each rollout is divided into four consecutive, non-overlapping step-index segments $Q_1 = [0, 50)$, $Q_2 = [50, 100)$, $Q_3 = [100, 150)$, and $Q_4 = [150, 200)$. $Q_1$ captures short-horizon accuracy, while $Q_4$ captures late-horizon error accumulation. For state coordinate $j \in \{\theta, \omega\}$ and segment $Q_q$, we compute

$$\text{NRMSE}_j^{(q)} = \frac{\sqrt{\frac{1}{|I_q|} \sum_{t \in I_q} (x_{t,j} - \hat{x}_{t,j})^2}}{\sqrt{\frac{1}{|I_q|} \sum_{t \in I_q} x_{t,j}^2} + \epsilon},$$

where $I_q$ is the time-index set of segment $Q_q$. All checkpoints and adaptation steps are selected using held-out zero-input and step-response validation rollouts. The final test set is used only for reporting. Table entries are reported as mean $\pm$ standard deviation across repeated runs.

**Quantitative results.** Table 2 report segmented NRMSE for $\theta$ and $\omega$, respectively. After validation selection, Scratch SK diverged in rollout evaluation across seeds. We therefore exclude it from the main quantitative tables. This failure indicates that, without meta-learning, the fragmented target data are insufficient to adapt the same SK architecture from a random initialization in this task. The results show that Fast Meta-ICI remains non-divergent over the full rollout horizon and achieves performance comparable to the Full-Data SK-ICI Reference under step-response tests, while a visible gap remains in the late zero-input segments.

To better interpret the segmented NRMSE values, we decompose them into segment-wise RMSE and signal RMS in Appendix B.2.4. This decomposition shows that the increasing zero-input NRMSE is mainly caused by the faster decay of the damped signal RMS than the absolute RMSE, while the step-response $\theta$ peak at $Q_2$ corresponds to the main transient region of the response.

Table 2: Segmented NRMSE for the angle state $\theta$ and angular-velocity state $\omega$ over 200-step rollouts. Zero-input and step-response tests are reported separately. Lower is better.

| State | Method | Zero-input tests | | | | Step-response tests | | | |
|---|---|---|---|---|---|---|---|---|---|
| | | $Q_1$ | $Q_2$ | $Q_3$ | $Q_4$ | $Q_1$ | $Q_2$ | $Q_3$ | $Q_4$ |
| $\theta$ | Full-Data SK-ICI Reference | $0.0642 \pm 0.0110$ | $0.1171 \pm 0.0264$ | $0.1623 \pm 0.0438$ | $0.1736 \pm 0.0439$ | $0.0341 \pm 0.0105$ | $0.0799 \pm 0.0251$ | $0.0589 \pm 0.0184$ | $0.0522 \pm 0.0162$ |
| | **Fast Meta-ICI (Ours)** | $0.1090 \pm 0.0556$ | $0.2845 \pm 0.1210$ | $0.3912 \pm 0.1972$ | $0.3856 \pm 0.1973$ | $0.0922 \pm 0.0182$ | $0.2394 \pm 0.1392$ | $0.2228 \pm 0.01362$ | $0.2243 \pm 0.1930$ |
| $\omega$ | Full-Data SK-ICI Reference | $0.0731 \pm 0.0139$ | $0.1116 \pm 0.0261$ | $0.1416 \pm 0.0357$ | $0.1744 \pm 0.0428$ | $0.0587 \pm 0.0276$ | $0.1600 \pm 0.0431$ | $0.2533 \pm 0.0760$ | $0.3618 \pm 0.0571$ |
| | **Fast Meta-ICI (Ours)** | $0.1335 \pm 0.0605$ | $0.2893 \pm 0.1279$ | $0.3679 \pm 0.1544$ | $0.4591 \pm 0.2419$ | $0.1284 \pm 0.0314$ | $0.3969 \pm 0.0673$ | $0.7468 \pm 0.1765$ | $0.9099 \pm 0.3482$ |

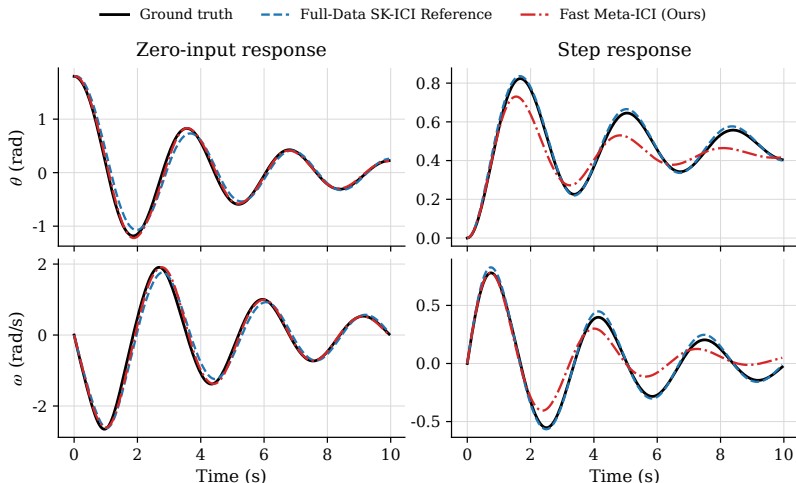

Figure 2: Representative 200-step open-loop rollouts for two selected test scenarios. The zero-input response uses $\theta_0 = 1.8$, $\omega_0 = 0$, and $r_t = 0$. The step response uses $x_0 = 0$ and constant step amplitude $r_t = 1.8$. The upper row shows the angle state $\theta(t)$, and the lower row shows the angular velocity state $\omega(t)$. Fast Meta-ICI is adapted from 200 scattered one-step target transitions and then evaluated on continuous long-horizon rollouts. The zero-input response tests autonomous dynamics recovery, while the step response evaluates forced dynamics under an input family not used during training.

**Representative rollouts.** To complement the segmented NRMSE values, Figure 2 reports two selected test scenarios from the final test set. The zero-input case uses $\theta_0 = 1.8$, $\omega_0 = 0$, and $r_t = 0$, while the step-response case uses $x_0 = 0$ and constant step amplitude $r_t = 1.8$. We use these rollouts as qualitative examples of the adapted model behavior. The main quantitative comparison is given by Table 2, which averages results over repeated runs and all test scenarios, whereas Figure 2 shows one selected case from each input family.

In this representative case, Fast Meta-ICI remains non-divergent over the full 200-step horizon under both input families and captures the dominant oscillatory behavior of the target system. The visible discrepancies reflect both the different state amplitudes and axis scales across panels and the fact that NRMSE is normalized segment-wise by the corresponding signal energy. Overall, the quantitative results show that Fast Meta-ICI approaches the Full-Data SK-ICI Reference on step-response tests, while a late-horizon gap remains on zero-input tests.

## 6 Conclusion

This work shows that closed-loop neural system identification can benefit from meta-learned stable internal operators. Instead of training each target system from scratch, Meta-ICI uses related closed-loop systems to provide an adaptive prior for the internal operator, making few-shot adaptation feasible under feedback-controlled data. A central outcome is that full-state observability makes the few-shot setting substantially easier to exploit. Fast Meta-ICI uses this structure so that adaptation can be performed with scattered one-step transitions, while the learned initialization is still selected for long-horizon rollout behavior. This is the main practical advantage of the method, since fragmented target data are common when long continuous target experiments are unavailable. The experiments support these conclusions in two complementary regimes. The Liénard benchmark shows that meta-learned internal operators can provide a useful prior when hidden dynamics require recurrent stable models. The pendulum benchmark shows that, when the state is fully observable, fragmented one-step target data can still guide a model toward reliable long-horizon counterfactual behavior. Future work should test this rollout-oriented prior on higher-dimensional controlled systems and examine its robustness to weaker task relatedness.

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

## A    Notation

For a sequence $x = \{x_t\}_{t \geq 0}$ with $x_t \in \mathbb{R}^{d_x}$, we write $x \in \ell_p^{d_x}$ if $\|x\|_{\ell_p} = \left(\sum_{t=0}^{\infty} \|x_t\|^p\right)^{1/p} < \infty, 1 \leq p < \infty$, and $x \in \ell_{\infty}^{d_x}$ if $\|x\|_{\ell_\infty} = \sup_{t \geq 0} |x_t| < \infty$. When the dimension is clear from context, we simply write $\ell_p$. An input-output operator $\Upsilon : \ell^{d_x} \to \ell^{d_y}$ maps an input sequence to an output sequence. It is causal if the output at time $t$ depends only on the input history up to time $t$, and strictly causal if the output at time $t$ depends only on the input history before time $t$. A causal operator $\Upsilon$ is $\mathcal{L}_p$-stable if $x \in \ell_p^{d_x} \implies \Upsilon(x) \in \ell_p^{d_y}$. It is incrementally $\mathcal{L}_p$-stable if there exists $\gamma \in [0, \infty)$ such that $\|\Upsilon(x) - \Upsilon(y)\|_{\ell_p} \leq \gamma \|x - y\|_{\ell_p}, \forall x, y \in \ell_p^{d_x}$. The sets of causal and strictly causal $\mathcal{L}_p$-stable operators are denoted by $\mathcal{L}_p^C$ and $\mathcal{L}_p^{SC}$, respectively.

## B    Detailed Experimental Settings

This appendix provides the experimental details omitted from the main text. We describe the system dynamics, data-generation procedures, model architectures, optimization settings, and evaluation protocols used in the two experiments. Appendix B.1 gives the details for the partially observable Liénard benchmark, and Appendix B.2 gives the details for the fully observable pendulum benchmark.

### B.1    Liénard System

This experiment evaluates Meta-ICI under partial observability. The benchmark uses a controlled Liénard-type nonlinear oscillator, where only the first state is measured and the second state remains hidden. We first describe the system dynamics and closed-loop data collection procedure, then specify the meta-learning data preparation, C-REN architecture, and optimization settings.

### B.1.1    System Dynamics and Data Generation

We use a controlled Liénard-type nonlinear oscillator as the partially observable benchmark (Strogatz, 2024; Khalil, 2002). Liénard systems form a classical family of two-dimensional nonlinear dynamical systems and include well-known nonlinear oscillators such as the van der Pol oscillator. The parameter $c$ controls the nonlinear damping strength, which makes the benchmark suitable for evaluating adaptation across related nonlinear systems.

The continuous-time target family is defined as follows:

$$\dot{x}_1 = x_2,$$
$$\dot{x}_2 = -c(1 + x_1^2)x_2 - x_1 + u,$$

where $c \sim \mathcal{U}[0.5, 2.0]$ parameterizes the source-task distribution. The target system uses $c = 1.57$. Only $x_1$ is measured, and $x_2$ is hidden from the model output. The closed-loop input is generated by a proportional controller

$$u_t = r_t - K_p(x_{1,t} + v_t),$$

where $K_p = 0.3 - 0.1c$ for each system and the controller output is clipped to $[-0.5, 0.5]$ before being added to the reference signal.

**Discretization.** The dynamics are numerically simulated using the standard fourth-order Runge-Kutta (RK4) integration method with a fixed time step of $\Delta t = 0.1$ s.

**Excitation signal.** The truncated colored Gaussian noise for the reference signal $r$ is generated by passing standard white noise through a second-order Butterworth low-pass filter with a cutoff frequency of 0.5 Hz. The filtered signal is z-score normalized, scaled to a standard deviation of 2.0, and clipped to the range $[-6.0, 6.0]$.

**Observation noise.** The zero-mean Gaussian observation noise $v$ has a standard deviation of $\sigma = 0.03$ and is truncated at $\pm 3\sigma$.

### B.1.2 Meta-Learning Data Preparation

We uniformly sample 100 source tasks from $c \in [0.5, 2.0]$. For each source task, we run closed-loop simulations to collect 200 independent trajectories, each consisting of 100 time steps. This offline dataset is used only during meta-training, where each task-level update samples disjoint support and query trajectory sets.

### B.1.3 C-REN Architecture

The stable operator $\boldsymbol{S}_\theta$ is parameterized by a C-REN with input dimension 1 and output dimension 1. Let $\xi_t \in \mathbb{R}^{n_\xi}$ denote the C-REN internal state and $w_t \in \mathbb{R}^q$ denote the nonlinear equilibrium variable. The recurrent update has the form

$$E\xi_{t+1} = F\xi_t + B_1 w_t + B_2 r_t,$$
$$y_t = C_2 \xi_t + D_{21} w_t,$$

where $w_t$ is computed through a lower-triangular implicit nonlinear block:

$$w_t = \tanh\left(\Lambda^{-1}(C_1 \xi_t + D_{11} w_t)\right).$$

In implementation, the components of $w_t$ are solved sequentially because $D_{11}$ is strictly lower triangular. The direct feedthrough terms from $r_t$ to $w_t$ and $y_t$ are omitted, enforcing strict causality with respect to the reference input. Following the C-REN parameterization, the matrices above are derived from a positive definite matrix $H = X^\top X + \epsilon I$ with $\epsilon = 10^{-3}$, which guarantees the contractive condition by construction. We set $n_\xi = 16$ and $q = 16$. The initial C-REN state is initialized from the first measured output using a learned linear map.

### B.1.4 Optimization and Baselines

**Meta-training.** The MAML inner update uses one gradient-descent step on the full support-set loss with adaptation rate $\alpha = 10^{-1}$, which is a deterministic update of the form $\theta_i' = \theta - \alpha \nabla_\theta \mathcal{L}_{\mathcal{T}_i}(\theta, \mathcal{D}_i^{sup})$. The outer loop uses Adam with meta-learning rate $\beta = 10^{-3}$. Each meta-update uses a task batch size of 8 source tasks. For each source task, the support set contains 5 trajectories and the query set contains 10 trajectories. Meta-training runs for 100 epochs.

Table 3: Extended data-size and adaptation-step sweep for Experiment 1.

| Method | Data Size | Adaptation Steps | | | | | | | | |
|---|---|---|---|---|---|---|---|---|---|---|
| | | 0 | 1 | 3 | 5 | 10 | 15 | 20 | 30 | 50 |
| **Full-Data ICI Reference** | 100 Trajs | | | | | $75.16 \pm 8.47$ | | | | |
| **Meta-ICI** (Ours) | 15 Trajs | $-73.26 \pm 20.52$ | $-30.71 \pm 15.67$ | $37.92 \pm 7.91$ | $59.61 \pm 11.59$ | $64.32 \pm 12.56$ | $65.80 \pm 12.45$ | $66.47 \pm 12.40$ | $66.96 \pm 12.40$ | $67.11 \pm 12.52$ |
| | 20 Trajs | | $-27.42 \pm 15.37$ | $43.57 \pm 8.32$ | $60.63 \pm 12.19$ | $64.45 \pm 12.65$ | $65.84 \pm 12.54$ | $66.46 \pm 12.50$ | $66.93 \pm 12.49$ | $67.09 \pm 12.60$ |
| | 30 Trajs | $-$ | $-24.26 \pm 14.53$ | $48.05 \pm 9.13$ | $59.66 \pm 12.68$ | $63.26 \pm 12.46$ | $65.12 \pm 12.12$ | $65.99 \pm 11.92$ | $66.65 \pm 11.73$ | $66.94 \pm 11.63$ |
| | 40 Trajs | | $-24.68 \pm 14.71$ | $47.43 \pm 8.97$ | $59.84 \pm 12.59$ | $63.52 \pm 12.46$ | $65.35 \pm 12.14$ | $66.22 \pm 11.94$ | $66.88 \pm 11.76$ | $67.18 \pm 11.68$ |
| | 50 Trajs | | $-22.99 \pm 14.39$ | $48.59 \pm 9.28$ | $59.57 \pm 12.60$ | $63.59 \pm 12.39$ | $65.53 \pm 12.06$ | $66.46 \pm 11.87$ | $67.16 \pm 11.73$ | $67.49 \pm 11.70$ |
| **Scratch ICI** | 15 Trajs | $-97.78 \pm 37.54$ | $-72.34 \pm 34.58$ | $-45.39 \pm 31.19$ | $-32.99 \pm 29.23$ | $-23.36 \pm 27.03$ | $-21.46 \pm 26.39$ | $-20.81 \pm 26.17$ | $-20.09 \pm 25.99$ | $-18.82 \pm 25.81$ |
| | 20 Trajs | | $-71.30 \pm 34.45$ | $-43.80 \pm 30.95$ | $-31.48 \pm 28.94$ | $-22.51 \pm 26.74$ | $-20.94 \pm 26.15$ | $-20.39 \pm 25.95$ | $-19.67 \pm 25.79$ | $-18.28 \pm 25.65$ |
| | 30 Trajs | $-$ | $-71.13 \pm 34.45$ | $-43.24 \pm 30.92$ | $-30.77 \pm 28.86$ | $-21.98 \pm 26.61$ | $-20.63 \pm 26.02$ | $-20.15 \pm 25.84$ | $-19.38 \pm 25.70$ | $-17.67 \pm 25.57$ |
| | 40 Trajs | | $-71.30 \pm 34.48$ | $-43.54 \pm 30.98$ | $-31.08 \pm 28.95$ | $-22.14 \pm 26.71$ | $-20.70 \pm 26.12$ | $-20.19 \pm 25.93$ | $-19.41 \pm 25.81$ | $-17.75 \pm 25.70$ |
| | 50 Trajs | | $-70.95 \pm 34.44$ | $-43.05 \pm 30.91$ | $-30.67 \pm 28.87$ | $-21.99 \pm 26.65$ | $-20.64 \pm 26.07$ | $-20.15 \pm 25.90$ | $-19.36 \pm 25.77$ | $-17.65 \pm 25.65$ |

Table 4: Long adaptation-step sweep for Experiment 1.

| Method | Data Size | Adaptation Steps | | | | | |
|---|---|---|---|---|---|---|---|
| | | 50 | 100 | 150 | 200 | 300 | 500 |
| **Full-Data ICI Reference** | 100 Trajs | | | $75.16 \pm 8.47$ | | | |
| **Meta-ICI** (Ours) | 5 Trajs | $66.97 \pm 12.64$ | $67.30 \pm 12.71$ | $67.71 \pm 12.68$ | $68.12 \pm 12.65$ | $68.78 \pm 12.63$ | $69.57 \pm 12.75$ |
| | 10 Trajs | $67.18 \pm 12.39$ | $67.33 \pm 12.66$ | $67.69 \pm 12.77$ | $68.08 \pm 12.83$ | $68.74 \pm 12.97$ | $69.61 \pm 13.29$ |
| | 50 Trajs | $67.49 \pm 11.70$ | $68.11 \pm 11.61$ | $68.70 \pm 11.40$ | $69.21 \pm 11.16$ | $70.05 \pm 10.75$ | $71.18 \pm 10.37$ |
| **Scratch ICI** | 5 Trajs | $-19.35 \pm 26.24$ | $-14.96 \pm 26.69$ | $-1.76 \pm 27.17$ | $33.88 \pm 22.47$ | $60.62 \pm 15.96$ | $64.89 \pm 17.42$ |
| | 10 Trajs | $-18.73 \pm 25.85$ | $-13.20 \pm 26.09$ | $5.31 \pm 26.08$ | $48.14 \pm 19.61$ | $62.78 \pm 16.93$ | $65.38 \pm 18.30$ |
| | 50 Trajs | $-17.65 \pm 25.65$ | $-9.01 \pm 25.88$ | $32.51 \pm 23.12$ | $63.14 \pm 13.82$ | $65.94 \pm 13.17$ | $68.86 \pm 12.84$ |

**Target adaptation.** After meta-training, Meta-ICI is adapted to the target system using the selected number of target trajectories in Table 1. Each adaptation step is a full-batch gradient descent update on the complete target adaptation set, with learning rate $3 \times 10^{-2}$.

**Baselines.** The **Full-Data ICI Reference** baseline uses the same C-REN architecture and is trained on 100 target-system trajectories with Adam, learning rate $10^{-3}$, batch size 25, and 100 epochs. The **Scratch ICI** baseline is randomly initialized and then adapted on the same target adaptation trajectories as Meta-ICI, using the same full-batch gradient descent update, learning rate, and adaptation-step budget.

### B.1.5 Extended Adaptation Sweep

To examine whether the gap between Meta-ICI and the Full-Data ICI Reference is caused by insufficient target data or insufficient adaptation steps, we conduct an extended adaptation sweep for Experiment 1. The results are reported in Tables 3 and 4. All values are FIT scores reported as mean ± standard deviation over 5 random seeds.

The extended sweep shows that Meta-ICI does not strictly plateau at 10 adaptation steps. However, the improvement becomes gradual, and the mean performance remains below the Full-Data ICI Reference. Scratch ICI also improves when both target data size and adaptation budget are increased. This indicates that the main advantage of Meta-ICI lies in rapid adaptation from a strong initialization.

These results clarify the practical limitation of the partially observable setting. Additional target data and longer adaptation can further improve performance, but the tested budgets do not fully close the mean gap to the full-data reference. Therefore, Meta-ICI should be interpreted as a data-efficient adaptation method, while abundant target-system trajectories can still provide additional information about target-specific hidden dynamics.

Table 5: Wall-clock timing for Experiment 1. Meta-training is an offline cost shared across target systems, whereas target adaptation is the online cost required for a new target system.

| Procedure | Experimental protocol | Wall-clock time |
|---|---|---|
| Meta-ICI meta-training | 100 meta-training epochs | 1914 s |
| Meta-ICI target adaptation | 1 target trajectory of length 100 | 0.95 s |
| Full-Data ICI Reference training | 100 target trajectories, 100 epochs | 427 s |

### B.1.6 Timing Analysis

We report wall-clock timing results for Experiment 1 in Table 5. All timings were measured on the same laptop workstation with an Intel Core i7-13700H CPU and an NVIDIA RTX A1000 Laptop GPU with 6GB memory, running Kubuntu 24.04.4. All neural-network training and adaptation runs were executed on the GPU. The timings are measured under the actual experimental protocols used to obtain the reported results. They include only training or adaptation time and exclude data collection time. Meta-training is an offline cost shared across target systems, whereas target adaptation is the online cost required for a new target system. For target adaptation, the measured time varied between 0.85 s and 0.95 s across repeated runs, and we report the maximum observed value.

The timing results show that meta-training is the main offline cost of Meta-ICI. Once the meta-learned initialization is available, adapting to a new target system is lightweight, requiring at most 0.95 s in this experiment. Thus, Meta-ICI provides a practical advantage at the target-adaptation stage, while the offline meta-training cost is higher than training a single full-data reference model on the target system.

## B.2 Pendulum System

This experiment evaluates Fast Meta-ICI in the fully observable setting. The benchmark uses a damped nonlinear pendulum with full-state measurements, allowing target adaptation to use scattered one-step transitions. We describe the closed-loop data collection procedure, the shifted test protocol, the validation and reporting rules, and the Schur-Koopman implementation details.

### B.2.1 System Dynamics and Closed-Loop Data Collection

We use a damped nonlinear pendulum as the fully observable benchmark. Pendulum systems are standard nonlinear mechanical benchmarks in dynamics and control because the restoring term $sin(\theta)$ induces strong state-dependent nonlinearity while the model remains low-dimensional and interpretable (Boubaker, 2013; Strogatz, 2024). The dynamics of the damped nonlinear pendulum with state $x = [\theta, \omega]^\top$ and scalar input $u$ is given as follows:

$$\dot{\theta} = \omega, \qquad \dot{\omega} = -\Omega^2 \sin(\theta) - 2\zeta\Omega\omega + u.$$

The target system uses $\zeta = 0.1$ and $\Omega = 2.0$. All trajectories are simulated with RK4 integration and a time step of $\Delta t = 0.05$. The closed-loop data are generated using a negative-feedback adaptive PD controller

$$u_t = r_t - K(x_t), \qquad K(x_t) = (2\Omega + 1)\theta_t + 2\zeta\omega_t.$$

For the target system parameters, this yields $K(x_t) = 5\theta_t + 0.2\omega_t$. Observation noise is zero-mean Gaussian noise with standard deviation 0.01, truncated at $\pm 0.03$.

**Training excitation.** Both target-system full-data training and meta-training use randomized bounded multisine references. Each reference is synthesized from 30 sinusoidal components with frequencies uniformly covering 0.5–10 Hz, random phases, and trajectory-wise random amplitude bounds. The target full-data set contains 1600 trajectories of length 50; the sequence loss during full-data training uses a horizon of 10. For meta-training, we sample 64 tasks with $\zeta \in [0.05, 0.3]$ and $\Omega \in [0.5, 4.5]$. Each task contains 1800 trajectories of length 50. Initial angles are sampled uniformly from $[-3, 3]$ and initial angular velocities are set to zero.

### B.2.2 Test Protocol and Validation Selection

The test inputs are deliberately different from the multisine training excitation. We evaluate two input families consisting of zero-input free responses and step responses. The zero-input tests use initial angles

$$\theta_0 \in \{0.5, 1.0, 1.8, 2.5, 2.8\}, \qquad \omega_0 = 0,$$

with $r_t = 0$. The step-response tests use $x_0 = 0$ and step amplitudes

$$r_t \in \{0.5, 1.0, 1.8, 2.4, 2.9\}.$$

All reported test rollouts use a horizon of 200 steps.

To avoid selecting checkpoints or adaptation steps on the test set, we use a held-out validation protocol with intermediate zero-input and step-response magnitudes. The validation set contains zero-input tests with $\theta_0 \in \{0.75, 1.5, 2.2\}$ and step-response tests with amplitudes $\{0.75, 1.5, 2.6\}$. All validation rollouts use a horizon of 200 steps. The validation score is the average late-horizon NRMSE over the validation rollouts and over the two state coordinates. Each 200-step rollout is split into four equal segments $Q_1$–$Q_4$, each containing 50 steps. For Fast Meta-ICI, the final test metrics are computed over five independent adaptation-data seeds. For each seed, the model is adapted from $K = 200$ randomly sampled scattered single-step transitions from the target system. For the Full-Data SK-ICI Reference, we train five full-data models and retain the best checkpoint of each seed according to the same validation score, evaluated after every training epoch. For Fast Meta-ICI, each reported table entry is computed by first averaging over the five test scenarios within the same input family, then reporting the mean $\pm$ standard deviation over the five adaptation-data seeds. For the Full-Data SK-ICI Reference, the tables report the mean $\pm$ standard deviation over the five validation-selected full-data runs.

### B.2.3 Schur-Koopman Architecture and Optimization

The SK model uses a 32-dimensional latent state. The encoder and decoder are three-layer bias-free MLPs with hidden dimension 64 and Tanh activations. The latent dynamics matrix is parameterized through a Schur-style stable linear map with maximum spectral radius 0.99. The control input is injected through a learned bias-free linear map from the scalar input to the two-dimensional state space. The reconstruction loss weight is set to 1.0.

**Full-Data SK-ICI Reference.** The full-data reference model is trained on the target system with Adam, learning rate $10^{-3}$, batch size 256, and 300 epochs. We do not use a fixed final epoch. After every training epoch, we evaluate the model on the held-out validation scenarios and retain the best checkpoint for each training seed. The resulting validation-selected full-data models are used only as an empirical reference under the same SK architecture and evaluation protocol.

**Fast Meta-ICI.** Hybrid MAML uses one inner-loop step, inner-loop learning rate $10^{-3}$, meta learning rate $5 \times 10^{-4}$, task batch size 32, query batch size 64, and outer-loop rollout horizon 10. The meta-training process runs for up to 30000 outer-loop updates, with coarse validation-based checkpoint selection from checkpoints evaluated every 100 outer-loop updates. Meta-SGD is enabled for the SK Hybrid MAML run, with initial per-parameter learning rate $10^{-3}$ (Li et al., 2017). At each validation point, the current checkpoint is adapted on a fixed validation adaptation set and evaluated across all candidate adaptation steps from 0 to 19. The checkpoint and adaptation step with the lowest validation score are selected before final testing.

**Target adaptation.** Fast Meta-ICI is adapted with the learned Meta-SGD per-parameter learning rates from the selected checkpoint. The selected adaptation step is reused across the five adaptation-data seeds in final testing. For Scratch SK, we first select the adaptation step on the validation set using the same candidate steps from 0 to 19. The selected Scratch SK step is then reused across the five adaptation-data seeds. Under the current configuration, Scratch SK rollouts diverged across seeds and are therefore excluded from the main table.

Table 6: Diagnostic decomposition of the segmented NRMSE in Table 2. $Z$-$Q_1$–$Z$-$Q_4$ denote zero-input segments, and $S$-$Q_1$–$S$-$Q_4$ denote step-response segments. The signal RMS is computed from the test trajectories and is shared by both methods.

| Quantity | State | Model / Source | $Z$-$Q_1$ | $Z$-$Q_2$ | $Z$-$Q_3$ | $Z$-$Q_4$ | $S$-$Q_1$ | $S$-$Q_2$ | $S$-$Q_3$ | $S$-$Q_4$ |
|---|---|---|---|---|---|---|---|---|---|---|
| RMSE | $\theta$ | Full-Data SK-ICI Ref. | 0.0962 | 0.0850 | 0.0613 | 0.0404 | 0.0212 | 0.0467 | 0.0363 | 0.0273 |
| RMSE | $\theta$ | Fast Meta-ICI | 0.1413 | 0.1839 | 0.1361 | 0.0774 | 0.0591 | 0.1580 | 0.1584 | 0.1437 |
| Signal RMS | $\theta$ | Test data | 1.0857 | 0.5417 | 0.3197 | 0.1964 | 0.6024 | 0.4334 | 0.5023 | 0.4695 |
| RMSE | $\omega$ | Full-Data SK-ICI Ref. | 0.1317 | 0.1433 | 0.1160 | 0.0741 | 0.0311 | 0.0690 | 0.0612 | 0.0494 |
| RMSE | $\omega$ | Fast Meta-ICI | 0.2285 | 0.3512 | 0.2654 | 0.1708 | 0.0674 | 0.1570 | 0.1761 | 0.1144 |
| Signal RMS | $\omega$ | Test data | 1.4405 | 1.0343 | 0.6357 | 0.3836 | 0.4710 | 0.3251 | 0.1829 | 0.1109 |

Table 7: Wall-clock timing for Experiment 2. Meta-training is an offline cost shared across target systems, whereas target adaptation is the online cost required for a new target system.

| Procedure | Experimental protocol | Wall-clock time |
|---|---|---|
| Fast Meta-ICI meta-training | 30000 outer-loop updates | 1214 s |
| Fast Meta-ICI target adaptation | 9 adaptation steps on 200 scattered transitions | 0.02 s |
| Full-Data SK-ICI Reference training | 300 epochs | 530 s |

### B.2.4 Diagnostic Decomposition of Segmented NRMSE

The NRMSE values reported in Table 2 are computed separately over four non-overlapping 50-step segments. To explain the non-monotonic patterns observed across these segments, we decompose the NRMSE into its numerator, the segment-wise RMSE, and its denominator, the segment-wise signal RMS. This diagnostic is computed from the same rollouts used in Table 2. The signal RMS is identical for the compared models because they are evaluated on the same test scenarios. The values in Table 6 are means computed over the same rollouts used for Table 2. Here we report the mean RMSE and RMS decomposition only.

In the zero-input responses, the pendulum dynamics are damped, and the signal amplitude decreases over time. For example, for the Full-Data SK-ICI Reference on the zero-input angle response, the RMSE decreases from 0.0962 in $Q_1$ to 0.0404 in $Q_4$, whereas the signal RMS decreases from 1.0857 to 0.1964. Thus, the NRMSE can increase even when the absolute prediction error decreases, because the denominator decreases faster than the numerator. For the step-response angle state, the $Q_2$ peak has a different origin. The RMSE itself peaks in $Q_2$, corresponding to the main transient part of the response, and the smaller signal RMS in $Q_2$ further increases the normalized error. This explains the non-monotonic pattern in the step-response angle NRMSE. Moreover, the Full-Data SK-ICI Reference generally shows a faster RMSE decay in the zero-input autonomous response, which is expected because it is trained with substantially richer target-system trajectories and can better fit the target-specific damping and natural response. Fast Meta-ICI remains non-divergent and comparable in many segments, but its slower RMSE decay and larger variability suggest residual phase or damping mismatch in long-horizon autonomous rollouts.

### B.2.5 Timing Analysis

We report wall-clock timing results for the main training and adaptation stages in Table 7. The timings are measured under the actual experimental protocols used to obtain the reported results. They include only training or adaptation time and exclude data collection time. Meta-training is an offline cost shared across target systems, whereas target adaptation is the online cost required for a new target system. These values are intended as practical reference timings for each stage, rather than as a parameter-matched computational complexity benchmark.

---

**Algorithm 1** Hybrid MAML procedure used in Experiment 2

---

1: **Input** source tasks, validation set, target adaptation set
2: Initialize meta-parameter $\theta$ and best validation score $J_{\text{best}} \leftarrow +\infty$
3: **for** outer-loop iteration $m = 1, \ldots, M$ **do**
4:     Sample a task batch $\mathcal{B}$
5:     **for** each task $\mathcal{T}_i \in \mathcal{B}$ **do**
6:         Sample scattered support transitions $\mathcal{D}_i^{\text{sup}} = \{(x_t, r_t, x_{t+1})\}$
7:         Compute the support loss by the step-wise loss in equation 12
8:         The reconstruction term is computed by

$$z = \text{detach}(\phi_\theta(x)), \quad x_{\text{rec}} = \psi_\theta(z), \quad \hat{z} = \phi_\theta(x_{\text{rec}}), \quad \mathcal{L}_{\text{rec}} = \text{mean}\|z - \hat{z}\|^2.$$

9:         Obtain the adapted parameter $\theta_i'$ by the inner update
10:        Sample contiguous query rollout segments $\mathcal{D}_i^{\text{qry}}$
11:        Compute the rollout prediction error by Algorithm 2
12:        Compute the query loss by the rollout loss in equation 13, including $\mathcal{L}_{\text{rec}}(\theta_i', \mathcal{D}_i^{\text{qry}})$
13:     **end for**
14:     Update $\theta$ using the average query loss over $\mathcal{B}$
15:     **if** $m \bmod 100 = 0$ **then**
16:         Evaluate the current checkpoint by Algorithm 3
17:         **if** the returned validation score is lower than $J_{\text{best}}$ **then**
18:             Save the current checkpoint and the returned adaptation step
19:         **end if**
20:     **end if**
21: **end for**
22: Adapt the selected checkpoint on $\mathcal{D}_\star^{\text{adapt}}$ using the selected adaptation step
23: **return** return the adapted parameter for the current target adaptation set

---

The timing results show that meta-training is the main offline cost of Fast Meta-ICI. Once the meta-learned initialization is available, adapting to a new target system is lightweight, requiring only 0.02 s in this experiment. This is the computational sense in which Fast Meta-ICI is fast at target adaptation. In addition, it can use fragmented one-step target data rather than long contiguous target trajectories.

### B.2.6   Hybrid-loss computation and validation selection.

To make the implementation of the hybrid objective explicit, we provide the pseudocode used in Experiment 2 (see Algorithm 1). The support loss is computed on scattered one-step transitions and is used for the inner adaptation update. The query loss is computed on contiguous rollout segments and is used for the outer meta-update. During meta-training, every 100 outer-loop updates, the current checkpoint is evaluated on the validation set with candidate adaptation steps from 0 to 19. The checkpoint and adaptation step with the best validation score are selected for final testing.

## C   Proof of $\mathcal{L}_p$ Stability for Schur-Koopman Architecture

This section provides the detailed proof of Theorem 1.

Define $z_t = \phi_\theta(x_t)$ and we have:

$$
\begin{aligned}
z_{t+1} &= \phi_\theta(x_{t+1}) \\
&= \phi_\theta\big(\psi_\theta(A_\theta \phi_\theta(x_t)) + g_\theta(x_t) r_t\big) \\
&= \phi_\theta\big(\psi_\theta(A_\theta z_t) + g_\theta(x_t) r_t\big)
\end{aligned}
\tag{14}
$$

---

**Algorithm 2** Rollout prediction error

1: **Input** adapted parameter $\theta_i'$, query rollout segments $\mathcal{D}_i^{\text{qry}}$
2: Initialize the accumulated error $E \leftarrow 0$
3: **for** each segment $(x_t, r_{t:t+T-1}, x_{t+1:t+T}) \in \mathcal{D}_i^{\text{qry}}$ **do**
4:     Set $\hat{x}_t = x_t$
5:     **for** $h = 0, \ldots, T-1$ **do**
6:         Predict
$$\hat{x}_{t+h+1} = f_{\theta_i'}(\hat{x}_{t+h}, r_{t+h})$$
7:         Accumulate
$$E \leftarrow E + \|x_{t+h+1} - \hat{x}_{t+h+1}\|^2$$
8:     **end for**
9: **end for**
10: Normalize $E$ over the query rollout segments
11: **return** $E$

---

**Algorithm 3** Validation selection of checkpoint and adaptation step

1: **Input** checkpoint $\theta$, validation adaptation set, validation rollout set
2: Initialize $J_{\text{ckpt}} \leftarrow +\infty$
3: **for** $s = 0, \ldots, 19$ **do**
4:     Adapt the checkpoint for $s$ steps on the validation adaptation set
5:     Evaluate the adapted model on the validation rollout set
6:     Compute the validation score used in Appendix B.2.2
7:     **if** the score is lower than $J_{\text{ckpt}}$ **then**
8:         Save this score and step $s$
9:     **end if**
10: **end for**
11: **return** the best validation score and the corresponding adaptation step

---

Rewrite equation 14 as:

$$
\begin{aligned}
z_{t+1} &= \phi_\theta\big(\psi_\theta(A_\theta z_t)\big) + \Big[\phi_\theta\big(\psi_\theta(A_\theta z_t) + g_\theta(x_t)r_t\big) - \phi_\theta\big(\psi_\theta(A_\theta z_t)\big)\Big] \\
&= F(z_t) + w_t
\end{aligned}
\tag{15}
$$

where $w_t = \phi_\theta\big(\psi_\theta(A_\theta z_t) + g_\theta(x_t)r_t\big) - \phi_\theta\big(\psi_\theta(A_\theta z_t)\big)$.

According to the Lipschitz of $\phi$, we have

$$\|w_t\|_2 \leq L_\phi \|g_\theta(x_t)r_t\|_2 \leq L_\phi \Gamma \|r_t\|_2$$

and by the norm equivalence, we can get

$$\|w_t\|_P \leq c_2 \|w_t\|_2 \leq c_2 L_\phi \Gamma \|r_t\|_2$$

where $c_2 > 0$.

With equation 15, we have:

$$\|z_{t+1}\|_P \leq \lambda \|z_t\|_P + \|w_t\|_P \tag{16}$$

Expanding the inequality 16, we have:

$$\|z_t\|_P \leq \sum_{k=0}^{t-1} \lambda^{t-1-k} \|w_k\|_P$$

By Young's inequality for convolution, we have:

$$\|\boldsymbol{z}\|_{\ell_p,P} \le \frac{1}{1-\lambda}\|\boldsymbol{w}\|_{\ell_p,P} \le \frac{c_2 L_\phi \Gamma}{1-\lambda}\|\boldsymbol{r}\|_{\ell_p} \tag{17}$$

where $\|\boldsymbol{z}\|_{\ell_p,P} = \|(\|z_0\|_P, \|z_1\|_P, ...)\|_{\ell_p}$.

Applying the Euclidean norm to both sides of equation 8, we have:

$$\begin{aligned}\|x_{t+1}\|_2 &\le L_\psi \|A_\theta z_t\|_2 + \Gamma\|r_t\|_2 \\ &\le \frac{L_\psi \|A_\theta\|_2}{c_1}\|z_t\|_P + \Gamma\|r_t\|_2\end{aligned}$$

where $c_1 > 0$.

Let $\nu = \frac{L_\psi \|A_\theta\|_2}{c_1}$. Therefore, we have:

$$\left(\sum_{t=0}^{\infty}\|x_{t+1}\|_2^p\right)^{1/p} \le \left(\sum_{t=0}^{\infty}(\nu\|z_t\|_P + \Gamma\|r_t\|_2)^p\right)^{1/p}$$

By Minkowski's Inequality, we have:

$$\left(\sum_{t=0}^{\infty}(\nu\|z_t\|_P + \Gamma\|r_t\|_2)^p\right)^{1/p} \le \left(\sum_{t=0}^{\infty}(\nu\|z_t\|_P)^p\right)^{1/p} + \left(\sum_{t=0}^{\infty}(\Gamma\|r_t\|_2)^p\right)^{1/p}$$

Therefore, we have

$$\|\boldsymbol{x}\|_{\ell_p} \le \nu\|\boldsymbol{z}\|_{\ell_p,P} + \Gamma\|\boldsymbol{r}\|_{\ell_p} \tag{18}$$

Substituting equation 17 into equation 18, we have:

$$\|\boldsymbol{x}\|_{\ell_p} \le \left(\frac{c_2 \nu L_\phi \Gamma}{(1-\lambda)} + \Gamma\right)\|\boldsymbol{r}\|_{\ell_p}$$

Let $C = \frac{c_2 \nu L_\phi \Gamma}{(1-\lambda)} + \Gamma$. Obviously, $C > 0$, and therefore we have:

$$\|\boldsymbol{x}\|_{\ell_p} \le C\|\boldsymbol{r}\|_{\ell_p}$$

