# OpenReview forum: "Few-Shot Closed-Loop Neural System Identification via Meta-Learning"
_TMLR — Under review for TMLR_

### Review · Reviewer_vA3m · 2026-06-17

**Summary Of Contributions:**

This paper studies few-shot closed-loop neural system identification within a meta-learning framework. The motivation is that many physical systems can only be excited under feedback control, and identification data for a new target system is often scarce, making it hard to recover unbiased open-loop dynamics from limited closed-loop data. To my knowledge, no prior work explicitly combines meta-learning with closed-loop identification while correcting for feedback-induced bias, so this problem formulation is a useful contribution.
The authors extend the existing Internal Controller Identification (ICI) framework into a meta-learning setting (Meta-ICI), meta-learning the initialization of an ICI-compatible stable internal operator across related closed-loop source systems. For fully observable systems, they further propose Fast Meta-ICI, exploiting the Markov property of the state to allow adaptation from fragmented one-step transitions while still meta-optimizing for long-horizon rollout accuracy, instantiated via a Schur-Koopman (SK) operator with a built-in spectral-radius constraint and an accompanying Lp-stability theorem. Experiments on Liénard and pendulum systems show clear utility of the proposed methods, though some results are not fully explained.

Key strengths:
- Novel and well-motivated problem formulation, addressing a genuine gap between closed-loop identification and meta-learning literatures
- Clear empirical utility: Meta-ICI substantially outperforms scratch training using the amount of trajectories, and Fast Meta-ICI remains non-divergent over long horizons where a non-meta-learned baseline diverges
- A theoretical Lp-stability guarantee for the proposed SK architecture

Weaknesses:
- Related work section omits several directly relevant works, particularly on spectral-radius-constrained Koopman operators developed outside the meta-learning setting, with no discussion or comparison
- Inconsistent notation throughout, and the ICI formulation being extended is not explained in enough detail for a reader unfamiliar with the original paper
- Several experimental results show unexplained or counterintuitive patterns that are not discussed
- SK parameterization is not supported by ablation studies
- Higher-dimensional benchmark. Both experiments use 2D state spaces. Even a modest extension to a 4–6 dimensional system (e.g., a coupled pendulum or a simple robot arm) would significantly strengthen the scalability claim in the conclusion.

**Additional Comments:**

One question I would like the authors to address: in equation (8), the reference signal r_t enters only linearly through the term g_θ(x_t)r_t, while the autonomous part of the dynamics is captured entirely by the latent linear evolution. This means the model's Koopman-style lifting only applies to the autonomous part of the dynamics, while the control channel is handled in a structurally restricted way that cannot capture nonlinear interactions between the input and the lifted state representation, is this not a limitation on the generality of the approach? I can see that this choice is convenient and presumably sufficient for the pendulum benchmark considered here, but I would like to understand from the authors how general this formulation is expected to be for closed-loop identification problems more broadly, and whether systems where the input's effect should also be subject to the nonlinear lifting (rather than entering affinely in the original state coordinates) would be properly handled by this architecture.

**Audience:**

Yes

**Audience Explanation:**

I believe a meaningful part of TMLR's audience, in particular researchers in the control and dynamical systems community working with neural network models, would find this paper interesting. It is, to my knowledge, among the first papers to combine meta-learning with closed-loop system identification, which is a useful and underexplored direction.

**Broader Impact Concerns:**

I do not have specific broader impact concerns about this work. The paper addresses a methodological problem in system identification for controlled dynamical systems, and the stability guarantees built into the proposed architecture are, if anything, a safety-positive design choice for downstream control applications. I do not believe a dedicated Broader Impact Statement is necessary for this submission.

**Claims And Evidence:**

Yes

**Claims Explanation:**

I think most of the individual claims made throughout the paper are reasonably convincing when read locally, and the overall narrative is coherent. The central claims, that Meta-ICI improves few-shot adaptation over scratch training with the same number of trajectories, and that Fast Meta-ICI yields non-divergent long-horizon rollouts from fragmented data where a non-meta-learned baseline diverges, are clearly supported by the reported results, particularly the Scratch ICI/SK comparisons.
That said, I do not think the current evidence is as fully convincing as it could be. Several results in the experimental section show patterns that are not explained, for example, segments where the proposed method outperforms the full-data reference unexpectedly, or where error does not increase monotonically with the rollout horizon as one might expect, and the paper does not investigate or discuss these patterns. This does not undermine the core claims, but it does limit how confidently I can assess the precise strength of the results. I believe that with some additional analysis, discussion, and possibly a few more targeted and complex experiments, the existing evidence could become considerably more convincing, also considering the question I posed to the authors in the additional comments below.

**Requested Changes:**

Major:
- Notation and clarity: The paper's notation has a number of inconsistencies that should be fixed before publication. D^sup_i and D^qry_i are used in Section 3 and Section 4 without ever being formally defined as a partition of one singular trajectory or sampling scheme over the trajectories collected for a task; this is only implicitly clarified in Appendix A.1.2. The trajectory index (i,j) is introduced without explanation of what varies across different j for the same task i (e.g., initial condition, noise realization, reference signal realization), this should be stated explicitly where the notation is introduced. N_i (the number of trajectories per task) is also never defined. In Section 4.1, the loss expression given just before equation (7) is written as a function of a generic dataset D, but D is never defined at that point, only D_i has been defined previously, and the text should clarify whether this loss is meant to be evaluated per-task on D_i or on a generic set. In Section 4.2.2, the reconstruction loss L_rec appears as a function of θ in the rollout/query loss L^roll_i, but I believe it should be a function of the adapted parameter θ'i, consistent with how the rollout error term itself depends on θ'i, please clarify or correct this. The notation also alternates inconsistently between Ĝ⋆ and Ĝ_θ'⋆ to refer to the same constructed open-loop model; please use one notation consistently throughout. I would ask the authors to do a full consistency pass on notation given the number of small issues of this kind.
- Related work: The Schur-Koopman operator is presented as a novel solution to enforce the spectral-radius constraint during unconstrained optimization, but the related work section does not discuss or cite prior work pursuing essentially the same goal. In particular, "Learning Stable Koopman Embeddings for Identification and Control" (Fan et al.) proposes an unconstrained parameterization of a Schur-stable Koopman transition matrix specifically to avoid the non-convex, constrained optimization that a direct spectral-radius constraint would otherwise require, and uses a reconstruction-type loss for its encoder/decoder pair that is conceptually similar to the L_rec term proposed here. I believe this is a directly relevant predecessor that should be cited and discussed, and ideally the paper should explain concretely why its own Schur-form, with-rotation-blocks, parameterization is preferable, and where possible, include an empirical comparison against it. There are also other works pursuing spectrally-constrained Koopman-style latent dynamics with similar motivations (e.g., orthogonal-diagonal-orthogonal factorizations with bounded spectral radius in recent time-series forecasting work) that should at least be acknowledged.
- ICI formultion: Section 4.1 essentially restates the ICI formulation from Boroujeni et al. in a meta-learning context, but I had to read the original ICI paper to fully understand what equation (5) and the associated theorem are actually saying and why they are useful. Given that this paper is explicitly built as an extension of ICI, I believe it should explain the ICI parameterization and the relevance of equation (5)-(6) in more detail and in more intuitive terms, rather than referring to "the relevance of equation 5 and equation 6" without unpacking what that relevance is. A figure summarizing the ICI interconnection (potentially adapted, with permission/citation, from the original paper, or redrawn) or adapting them in a meta-learning setting would also help readers unfamiliar with ICI follow the construction without consulting the original source.
- Meta-ICI results, missing parts: In Table 1, performance appears to plateau around 10 adaptation step without closing the gap to the Full-Data ICI Reference (75.16%). It is unclear from the current results whether Meta-ICI could eventually match the full-data reference with more trajectories or adaptation steps, or whether there is a fundamental limitation (e.g., model capacity, linearization bias) preventing this. I would ask the authors to clarify this point, ideally with additional data points and/or steps. Additionally, I would ask for a timing analysis: how does the inference/adaptation time of Meta-ICI compare to the training time of the Full-Data ICI Reference, and what is the actual computational cost of the meta-training phase itself? This information seems important for assessing the practical value, that is usually one of the main advantage of meta learning but is currently missing. Finally, I noticed in Appendix A.1.1 that the target system's parameter (c = 1.57) lies within the training range used to sample source tasks (c ∼ U[0.5, 2.0]), meaning the target task is an interpolation case within the training distribution rather than a genuinely unseen, out-of-distribution system. The main text's language (e.g., "unseen target system") could mislead readers into thinking this tests extrapolation. I would ask the authors to state explicitly in the main text that the target task is drawn from within the support of the source task distribution, since this affects how the few-shot results should be interpreted.
- Fast Meta-ICI results, missing parts: In Tables 2 and 3, Fast Meta-ICI sometimes outperforms the Full-Data SK-ICI Reference (e.g., in some zero-input segments), which is counterintuitive given that the reference is trained on substantially more data. I would ask the authors to discuss why this occurs. Separately, in the step-response results for θ (Table 2), the Full-Data SK-ICI Reference shows a clear non-monotonic spike at Q2 (0.32) before decreasing at Q3 (0.20) and Q4 (0.16), whereas I would expect NRMSE to increase roughly monotonically with horizon length due to error accumulation. This pattern is not discussed in the paper. I suspect this may be related to the NRMSE normalization, but this should be investigated and explained rather than left unaddressed, since it currently undermines confidence in the metric and the comparison.
- Scenarios and runs: The main text refers to "test scenarios" and "repeated runs" in Section 5.2 without defining what varies across them. I found the definition of the zero-input/step-response initial conditions and amplitudes only in Appendix A.2.2, and I am still not fully sure whether "repeated runs" refers to the 5 seeds, or something else, since the terminology is used somewhat differently for the two compared methods. I would ask the authors to define these terms explicitly in the main text where they are first used, rather than relying on the appendix.

These would strengthen the work:
- The related work section on stable neural operators should also mention L2-stable neural state-space models (e.g., "L2RU: a Structured State Space Model with prescribed L2-bound" Massai et al.), even though, as far as I understand, such models may not satisfy the contraction property required by this paper's framework and would therefore not be directly usable as a substitute for C-REN. I think it is still worth citing this as a relevant but excluded alternative, with a brief explanation of why it does not fit the required L^SC_p / contraction requirements of ICI.
- "Fast" Meta-ICI. It is unclear what "Fast" is meant to convey. Is the claim about wall-clock adaptation time (e.g., because the inner loop only requires a single forward pass per gradient step on isolated transitions, rather than unrolling through a sequence as in the recurrent Meta-ICI case), or about data efficiency (because fragmented data are cheaper/faster to collect than continuous trajectories), or both? No timing measurements are reported anywhere in the paper to substantiate either interpretation. I would ask the authors to state explicitly what "Fast" refers to and, ideally, provide a quantitative comparison (e.g., wall-clock adaptation time per gradient step, or time to collect equivalent amounts of fragmented vs. contiguous data) to support the claim.
- Clarity of hybrid loss. The description of how the query rollout loss is computed and used to select θ during meta-training (and how θ' is selected at test time across rollout positions/seeds) is not fully clear to me. I understood that, in the experiments, the rollout loss is computed by predicting forward from an initial state across the full horizon, this is averaged over rollout positions, and then mean/std are reported over seeds, but it would help to have an explicit, step-by-step description of this procedure (perhaps as pseudocode) in the main text or appendix. Moreover, including how the time index t is chosen during adaptation/evaluation and whether it is sampled at random or fixed, I didn't understand this role from the paper.
- The experiments use 5 random seeds. Given the relatively large standard deviations already visible in Tables 1–3, I believe a larger number of seeds (e.g., 15-20) would substantially strengthen the statistical reliability of the mean/std comparisons, especially for the claims about Fast Meta-ICI approaching full-data performance.

---

> ### Author Response · Authors · 2026-07-03
> **Responses to the comments from Reviewer vA3m (Part 1/5)**
>
> We sincerely thank the reviewer for the constructive feedback, which has significantly improved the clarity and empirical rigor of our work. Below, we address your detailed comments point by point.
>
> > 1. Notation and clarity
>
> The manuscript has been revised to resolve the notation inconsistencies:
> - Defined $N_i$, trajectory index $j$, and the trajectory-level split $\mathcal{D}_i = \mathcal{D}_i^{\mathrm{sup}} \cup \mathcal{D}_i^{\mathrm{qry}}$ when the task datasets are introduced.
> - Clarified that the generic dataset $\mathcal{D}$ in the loss definition denotes a task-specific dataset ($\mathcal{D}_i^{\mathrm{sup}}$, $\mathcal{D}_i^{\mathrm{qry}}$, or $\mathcal{D}_i$).
> - Corrected the rollout loss so the reconstruction loss is evaluated using the adapted parameter $\theta_i'$, consistent with the rollout prediction term.
> - Unified the notation for the constructed target open-loop model as $\hat G_\star$ throughout the manuscript.
>
> > 2. Related work on Schur-stable / Koopman operators
>
> The related work section has been expanded to acknowledge stable Koopman embeddings with unconstrained Schur-stable parameterizations, spectrally constrained Koopman-style latent dynamics, and $\mathcal{L}_2$-stable sequence models (e.g., L2RU).
>
> - **Comparison with L2RU:** L2RU provides an input-output $\mathcal{L}_2$ bound.
> This differs from ICI's stability requirement, which targets a strictly causal stable internal operator in $\mathcal{L}_p^{SC}$ for all $p\in[1,\infty]$.
> An $\mathcal{L}_2$ guarantee does not cover inputs that are not $\ell_2$-bounded (e.g., step signals) and does not directly provide the contraction property used by C-REN.
> L2RU is a relevant alternative sequence model but not a direct substitute for the ICI construction.
>
> - **Comparison with Fan et al. (2026):** Fan et al. propose a general unconstrained Schur-stable parameterization representing any matrix with a spectral radius smaller than one, which can serve as a drop-in replacement for our latent matrix $A_\theta$.
> Our parameterization has a different purpose.
> It is a structured real-Schur parameterization designed to be lightweight and to make damped oscillatory modes explicit.
>
> Standard real Schur forms allow both $1 \times 1$ real blocks and $2 \times 2$ complex-conjugate blocks.
> For simplicity, our implementation uses only $2 \times 2$ rotation-scaling blocks (if n_z is even).
> This choice is less general than a complete Schur-stable parameterization, since real eigenvalues are represented as paired modes.
> In our Koopman-style latent model, this restriction is mitigated by the overcomplete latent dimension and the encoder-decoder maps, which can suppress redundant latent coordinates.
>
> The practical advantage of the rotation-scaling form is that it makes the spectral-radius constraint simple and unconstrained.
> Each block has eigenvalues $\rho_k e^{± i\omega_k}$, and therefore it suffices to parameterize $\rho_k=\bar\rho\sigma(a_k)$ with $\bar\rho<1$.
> This avoids constrained optimization over a general $2 \times 2$ block while retaining explicit oscillatory latent modes.
> Because the encoder and decoder themselves are nonlinear and expressive, isolating the effect of the Schur block type through an ablation study is not straightforward.
> A diagonal latent transition may still partially recover oscillatory behavior through the learned encoder-decoder maps.
> We therefore revised the text to present the rotation-block form as a lightweight, oscillation-aware, stability-preserving design choice within the proposed Fast Meta-ICI framework.
>
>
> > 3. ICI formulation and explanatory figure
>
> We have revised Section 4.1 to make the construction more intuitive for readers unfamiliar with the original work:
> - Added a new explanatory figure in Section 4.1 (redrawn from Boroujeni et al., 2025) to illustrate the ICI interconnection visually.
> - Explicitly stated that Eq. (5) defines the interconnection between $S_i$ and $K_i$, while Eq. (6) defines the induced open-loop input-output map $\hat G_i(\hat u)=\hat y^\circ$.
> - Clarified the practical role of the ICI theorem more directly in the text.

---

> ### Author Response · Authors · 2026-07-03
> **Responses to the comments from Reviewer vA3m (Part 2/5)**
>
> > 4.1 Meta-ICI experimental results (Exp. 1)
>
> We have added an extended adaptation sweep for Exp. 1 to examine whether the gap to the Full-Data ICI Reference is due to insufficient target data or insufficient adaptation steps.
>
> **Extended data-size and adaptation-step sweep for Exp. 1.**
>
> |Method|Trajs|0|1|3|5|10|15|20|30|50|
> |---|---:|---:|---:|---:|---:|---:|---:|---:|---:|---:|
> |**Full-Data ICI Reference**|100|75.16±8.47|--|--|--|--|--|--|--|--|
> |**Meta-ICI (Ours)**|15|-73.26±20.52|-30.71±15.67|37.92±7.91|59.61±11.59|64.32±12.56|65.80±12.45|66.47±12.40|66.96±12.40|67.11±12.52|
> ||20|--|-27.42±15.37|43.57±8.32|60.63±12.19|64.45±12.65|65.84±12.54|66.46±12.50|66.93±12.49|67.09±12.60|
> ||30|--|-24.26±14.53|48.05±9.13|59.66±12.68|63.26±12.46|65.12±12.12|65.99±11.92|66.65±11.73|66.94±11.63|
> ||40|--|-24.68±14.71|47.43±8.97|59.84±12.59|63.52±12.46|65.35±12.14|66.22±11.94|66.88±11.76|67.18±11.68|
> ||50|--|-22.99±14.39|48.59±9.28|59.57±12.60|63.59±12.39|65.53±12.06|66.46±11.87|67.16±11.73|67.49±11.70|
> |**Scratch ICI**|15|-97.78±37.54|-72.34±34.58|-45.39±31.19|-32.99±29.23|-23.36±27.03|-21.46±26.39|-20.81±26.17|-20.09±25.99|-18.82±25.81|
> ||20|--|-71.30±34.45|-43.80±30.95|-31.48±28.94|-22.51±26.74|-20.94±26.15|-20.39±25.95|-19.67±25.79|-18.28±25.65|
> ||30|--|-71.13±34.45|-43.24±30.92|-30.77±28.86|-21.98±26.61|-20.63±26.02|-20.15±25.84|-19.38±25.70|-17.67±25.57|
> ||40|--|-71.30±34.48|-43.54±30.98|-31.08±28.95|-22.14±26.71|-20.70±26.12|-20.19±25.93|-19.41±25.81|-17.75±25.70|
> ||50|--|-70.95±34.44|-43.05±30.91|-30.67±28.87|-21.99±26.65|-20.64±26.07|-20.15±25.90|-19.36±25.77|-17.65±25.65|
>
> **Long adaptation-step sweep for Exp. 1.**
>
> |Method|Trajs|50|100|150|200|300|500|
> |---|---:|---:|---:|---:|---:|---:|---:|
> |**Full-Data ICI Reference**|100|75.16±8.47|--|--|--|--|--|
> |**Meta-ICI (Ours)**|5|66.97±12.64|67.30±12.71|67.71±12.68|68.12±12.65|68.78±12.63|69.57±12.75|
> ||10|67.18±12.39|67.33±12.66|67.69±12.77|68.08±12.83|68.74±12.97|69.61±13.29|
> ||50|67.49±11.70|68.11±11.61|68.70±11.40|69.21±11.16|70.05±10.75|71.18±10.37|
> |**Scratch ICI**|5|-19.35±26.24|-14.96±26.69|-1.76±27.17|33.88±22.47|60.62±15.96|64.89±17.42|
> ||10|-18.73±25.85|-13.20±26.09|5.31±26.08|48.14±19.61|62.78±16.93|65.38±18.30|
> ||50|-17.65±25.65|-9.01±25.88|32.51±23.12|63.14±13.82|65.94±13.17|68.86±12.84|
>
> These results clarify the trend.
> Meta-ICI does not strictly plateau at 10 adaptation steps.
> The results also show that Scratch ICI can partially catch up when both the target data size and the adaptation budget are increased.
> The extended sweep shows that additional target data and longer adaptation can further improve performance, while the tested budgets do not fully close the mean gap to the full-data reference.
>
> > 4.2 Exp. 1 time
>
> We have added timing analyses for both experiments in Appendix B.1.6.
> For Exp. 1, the timing results are shown below.
> The values are measured under the actual experimental protocols used in the paper.
> Meta-training is an offline cost shared across target systems, while target adaptation is the online cost for a new target system.
> For target adaptation, we report the maximum observed time among repeated runs (measured time varied between 0.85 s and 0.95 s).
>
> **Wall-clock timing for Exp. 1.**
>
> | Procedure| Experimental protocol| Wall-clock time |
> | ---| ---| --- |
> | Meta-ICI meta-training| 100 meta-training epochs| 1914 s|
> | Meta-ICI target adaptation| 1 target trajectory of length 100| 0.95 s|
> | Full-Data ICI Reference training| 100 target trajectories, 100 epochs| 427 s|
>
> The results show that Meta-ICI requires a larger offline meta-training cost than training the Full-Data ICI Reference once on the target system.
> After meta-training, however, adapting Meta-ICI to a new target system is lightweight, requiring at most 0.95 s in this experiment.
> Together with the extended adaptation sweep, this supports the intended interpretation that Meta-ICI provides fast target adaptation from limited target data, while the meta-training stage is an offline cost.
>
> > 4.3 Exp. 1 text's language
>
> We have revised the wording to describe the target system as an in-distribution target task.
> The text explicitly states that the target parameter $c=1.57$ lies in the same range $c\in[0.5,2.0]$ used for the source tasks, clarifying that this experiment does not claim out-of-distribution extrapolation.

---

> > ### Author Response · Authors · 2026-07-03
> > **Responses to the comments from Reviewer vA3m (Part 3/5)**
> >
> > > 5. Fast Meta-ICI experimental results (Exp. 2) and abnormal patterns
> >
> > We revisited this experiment and found that two issues should be separated:
> > (i) the occasional cases where Fast Meta-ICI appeared to outperform the Full-Data SK-ICI Reference, and (ii) the non-monotonic segmented NRMSE values.
> >
> > First, regarding the issue (i), we found that the original Full-Data SK-ICI Reference (trained for 100 epochs) was under-optimized.
> > We have updated the results by extending the full-data training budget to 300 epochs while maintaining the same validation-based checkpoint selection protocol.
> >
> > As shown in the updated tables below (Z denotes zero-input, S denotes step-response), the properly optimized Full-Data SK-ICI Reference now consistently outperforms Fast Meta-ICI across all segments, resolving the first anomaly.
> >
> > **Updated segmented NRMSE.**
> >
> > |State|Method|Z-Q1|Z-Q2|Z-Q3|Z-Q4|S-Q1|S-Q2|S-Q3|S-Q4|
> > |---|---|---:|---:|---:|---:|---:|---:|---:|---:|
> > |$\theta$|Full-Data SK-ICI Ref.|0.0642±0.0110|0.1171±0.0264|0.1623±0.0438|0.1736±0.0439|0.0341±0.0105|0.0779±0.0251|0.0589±0.0184|0.0522±0.0162|
> > ||Fast Meta-ICI|0.1090±0.0556|0.2845±0.1210|0.3912±0.1972|0.3856±0.1973|0.0922±0.0182|0.2394±0.1392|0.2228±0.1362|0.2243±0.1930|
> > |$\omega$|Full-Data SK-ICI Ref.|0.0731±0.0139|0.1116±0.0261|0.1416±0.0357|0.1744±0.0428|0.0587±0.0276|0.1600±0.0431|0.2533±0.0760|0.3618±0.0571|
> > ||Fast Meta-ICI|0.1335±0.0605|0.2893±0.1279|0.3679±0.1544|0.4591±0.2419|0.1284±0.0341|0.3969±0.0673|0.7468±0.1765|0.9099±0.3482|
> >
> > Second, regarding the issue (ii), the reported Q1--Q4 values are segment-wise errors over four non-overlapping windows, not cumulative errors over increasing horizons.
> > Therefore, strict monotonicity is not expected for this metric.
> >
> > To clarify this point, we decomposed the reported NRMSE values into segment-wise RMSE and segment-wise signal RMS:
> >
> > **Segmented RMSE and signal RMS.**
> >
> > |State|Item|Z-Q1|Z-Q2|Z-Q3|Z-Q4|S-Q1|S-Q2|S-Q3|S-Q4|
> > |---|---|---:|---:|---:|---:|---:|---:|---:|---:|
> > |$\theta$|Full-Data RMSE|0.0962|0.0850|0.0613|0.0404|0.0212|0.0467|0.0363|0.0273|
> > ||Fast Meta-ICI RMSE|0.1413|0.1839|0.1361|0.0774|0.0591|0.1580|0.1584|0.1437|
> > ||Signal RMS|1.0857|0.5417|0.3197|0.1964|0.6024|0.4334|0.5023|0.4695|
> > |$\omega$|Full-Data RMSE|0.1317|0.1433|0.1160|0.0741|0.0311|0.0690|0.0612|0.0494|
> > ||Fast Meta-ICI RMSE|0.2285|0.3512|0.2654|0.1708|0.0674|0.1570|0.1761|0.1144|
> > ||Signal RMS|1.4405|1.0343|0.6357|0.3836|0.4710|0.3251|0.1829|0.1109|
> >
> > In the zero-input responses, the pendulum dynamics are damped, causing the signal RMS (the denominator) to decay rapidly over time. The NRMSE increases because the signal energy decays faster than the absolute prediction error (RMSE).
> > For the step-response angle state, the NRMSE peak at $Q_2$ corresponds to the main transient region of the response. Here, the absolute RMSE inherently peaks, while the signal RMS remains relatively small, driving up the normalized error.
> >
> > We have clarified in the main text that segmented NRMSE is a scale-normalized relative prediction metric and referenced the decomposition diagnostic in Appendix B.2.4.

---

> ### Author Response · Authors · 2026-07-03
> **Responses to the comments from Reviewer vA3m (Part 4/5)**
>
> > 6. Additional 20-seed diagnostic.
> As suggested, we ran an additional 20-seed diagnostic for both experiments to improve statistical reliability.
>
> For Exp. 1, the 20-seed results show the same trend as the original 5-seed report.
> Meta-ICI consistently outperforms Scratch ICI across all target-data sizes and adaptation-step budgets.
> The main conclusion is unchanged.
> Meta-ICI provides a strong initialization and substantially improves early-stage adaptation under partial observability.
>
> **20-seed FIT results for Exp. 1.**
>
> |Method|Trajs|0|1|3|5|10|
> |---|---:|---:|---:|---:|---:|---:|
> |**Full-Data ICI Reference**|100|76.52±6.92|--|--|--|--|
> |**Meta-ICI**|1|-49.35±25.65|-27.99±21.91|7.44±15.55|33.63±11.72|59.95±12.12|
> ||3|--|-22.05±20.97|22.79±13.39|50.90±11.53|64.33±14.42|
> ||5|--|-15.08±19.58|38.85±11.21|59.65±12.59|64.87±14.42|
> ||8|--|-16.91±20.03|36.36±11.56|59.20±12.50|64.91±14.54|
> ||10|--|-15.20±19.66|40.08±11.21|60.12±12.59|64.90±14.13|
> |**Scratch ICI**|1|-75.77±34.13|-56.63±32.15|-35.32±30.89|-24.96±30.38|-16.05±29.34|
> ||3|--|-55.41±32.06|-33.49±30.78|-23.27±30.17|-15.12±28.89|
> ||5|--|-54.14±31.96|-31.31±30.68|-21.07±29.96|-13.70±28.34|
> ||8|--|-54.71±31.99|-31.85±30.66|-21.31±29.92|-13.63±28.18|
> ||10|--|-53.85±31.92|-30.65±30.56|-20.33±29.73|-13.27±27.92|
>
> For Exp. 2, increasing the adaptation-data seeds makes the Fast Meta-ICI results more stable, with standard deviations generally decreasing. The model remains non-divergent over all 200-step rollouts.
> The 20-seed diagnostic also reveals that the Full-Data SK-ICI Reference has non-negligible optimization sensitivity under its stochastic training protocol.
> In particular, a small number of full-data runs fail to optimize properly, which substantially inflates the reported mean and standard deviation..
> This confirms that the Full-Data SK-ICI Reference should be interpreted as an empirical reference under a specified training protocol, not as a theoretical upper bound.
>
> The 20-seed diagnostic results for Experiment 2 are shown below.
> Z-Q1--Z-Q4 denote the four zero-input segments, and S-Q1--S-Q4 denote the four step-response segments.
>
> **20-seed segmented NRMSE.**
>
> |State|Method|Z-Q1|Z-Q2|Z-Q3|Z-Q4|S-Q1|S-Q2|S-Q3|S-Q4|
> |---|---|---|---|---|---|---|---|---|---|
> |$\theta$|Full-Data|0.1768±0.3420|0.2427±0.3888|0.3596±0.6155|0.4728±0.9166|0.1559±0.3419|0.2173±0.3910|0.1965±0.3773|0.1832±0.3522|
> ||Fast Meta-ICI|0.1032±0.0417|0.2602±0.0826|0.3625±0.1316|0.3690±0.1404|0.0971±0.0245|0.1855±0.1131|0.1908±0.0971|0.1756±0.1405|
> |$\omega$|Full-Data|0.1655±0.2720|0.2057±0.2697|0.2400±0.2781|0.2909±0.2902|0.1595±0.2857|0.2715±0.2816|0.3778±0.2940|0.5359±0.3189|
> ||Fast Meta-ICI|0.1245±0.0454|0.2623±0.0896|0.3345±0.1104|0.4361±0.1688|0.1056±0.0282|0.3666±0.0615|0.6437±0.1426|0.7902±0.2515|
>
> We do not emphasize the 20-seed diagnostic in the main text. For Exp. 1, it confirms the reported data-efficiency trends. For Exp. 2, it primarily reveals rare stochastic optimization failures in the empirical Full-Data baseline, while our methods remain stable. Therefore, the main manuscript retains the properly optimized 5-seed (300-epoch) comparison, leaving a systematic robustness study of SK-operators to future work.
>
>
> > 7. Miscellaneous clarifications
>
> - **"Fast" & Timing:** We clarified that "Fast" refers to online target adaptation using scattered one-step transitions. As detailed in the new Appendix B.2.5, this adaptation takes only 0.02 s (compared to 530 s for full-data training).
> - **Scenarios & Runs:** Section 5.2 has been revised to explicitly define "test scenario" (fixed evaluation condition) and "repeated runs" (independent random seeds).
> - **Hybrid Loss & Time Index:** We added step-by-step pseudocode (Algorithm 1) in the Appendix B.2.6 and clarified that $t$ refers to a fixed discrete-time grid step, not a randomly sampled continuous variable.
> - **Higher-Dimensional Systems:** We agree that testing on higher-dimensional systems would further strengthen the evaluation. However, in the considered closed-loop setting, increasing the state dimension is not a simple simulator adjustment. It requires constructing a parameterized family of related systems, co-designing stabilizing controllers for all source and target tasks, and ensuring persistent closed-loop excitation across the distribution. These control-theoretic choices strongly influence identification outcomes. Introducing them without careful isolation could create confounding variables that obscure the core meta-learning contribution. We have explicitly added this as an important direction for future work in the revised conclusion.

---

> ### Author Response · Authors · 2026-07-03
> **Responses to the comments from Reviewer vA3m (Part 5/5)**
>
> > 8. Additional comments on input-affine SK architecture
>
> This is a good question.
> From a neural-network approximation perspective, if the true dynamics depend on nonlinear input terms such as $r_t^2$, $r_t^3$, or other nonlinear transformations of $r_t$, then Eq. (8) cannot represent such dependence exactly when only the raw input $r_t$ is used.
>
> Our view is that Eq. (8) should be interpreted as an input-affine model with respect to the chosen input coordinate.
> In control-oriented system identification, the input coordinate is often selected or transformed using prior knowledge.
> For example, if the relevant actuation variable scales as $r_t^2$ or $r_t^3$, one may define a transformed input such as $a_t=r_t^2$, $a_t=r_t^3$, or use an augmented input vector $a_t=[r_t,r_t^2,r_t^3]$, and then apply the same input-affine structure with respect to $a_t$.
>
> A practical example is wind-turbine drivetrain modeling.
> If wind speed $v$ is used directly as the raw external input, the drivetrain model would take the form
>
> $$
> \dot x = f(x,u,v).
> $$
>
> Here, $x$ denotes the drivetrain state and $u$ denotes the control input.
> The difficulty is that the wind-speed effect enters the drivetrain dynamics through aerodynamic power and aerodynamic torque.
> A common aerodynamic power relation is
>
> $$
> P_r=\frac{1}{2}\rho A C_p(\lambda,\beta)v^3,
> $$
>
> where $P_r$ is aerodynamic power, $\rho$ is air density, $A$ is rotor swept area, $C_p$ is the power coefficient, $\lambda$ is the tip-speed ratio, and $\beta$ is the blade pitch angle.
> The corresponding aerodynamic torque can be written as
>
> $$
> T_r=\frac{P_r}{\omega_r}=\frac{1}{2}\rho A\frac{C_p(\lambda,\beta)}{\omega_r}v^3,
> $$
> where $T_r$ is aerodynamic torque and $\omega_r$ is rotor speed.
> Thus, using $v$ directly requires the model to represent the cubic wind-speed dependence.
>
> In a control-oriented formulation, one can instead introduce $T_r$ as an intermediate input.
> For example, with a simplified drivetrain state and control input
> $$
> x=[\theta,\omega_r,\omega_g,T_g,\beta],
> \qquad
> u=[T_{g,\mathrm{ref}},\beta_{\mathrm{ref}}],
> \qquad
> r=T_r,
> $$
> the drivetrain dynamics can be written schematically as
> $$
> \dot x = F(x,u,T_r).
> $$
> Here, $\theta$ is the drivetrain torsion angle, $\omega_g$ is generator speed, $T_g$ is generator torque, $T_{g,\mathrm{ref}}$ is the generator-torque command, and $\beta_{\mathrm{ref}}$ is the pitch-angle command.
> In this coordinate, the nonlinear wind-speed effect is handled by the aerodynamic map from $v$ to $T_r$, while $T_r$ enters the drivetrain dynamics as a torque input.
> This is consistent with the modeling idea in [1], where wind-turbine dynamics are coupled with torque and pitch control.
>
> This example clarifies the limitation of the current SK instantiation.
> The input-affine structure is restrictive when applied directly to arbitrary raw inputs.
> In control-oriented identification, however, one can often choose a physically meaningful input coordinate or intermediate variable, such as $T_r$, so that the remaining dynamics are closer to the input-affine form.
> This structure can also keeps the stability argument tractable.
>
> ---
>
> [1] J. Xie, H. Dong, and X. Zhao, “Data-driven torque and pitch control of wind turbines via reinforcement learning,” Renewable Energy, vol. 215, 118893, 2023.

---

### Review · Reviewer_5pr5 · 2026-06-19

**Summary Of Contributions:**

The authors propose Meta-ICI, a meta-learning framework designed for few-shot closed-loop neural system identification. The approach aims to recover open-loop dynamics from limited feedback-controlled data by learning a transferable initialization for an Internal Controller Identification (ICI) operator across related source systems. For fully observable systems, the framework is extended to Fast Meta-ICI, which uses a Schur-Koopman architecture to enable adaptation from scattered, one-step transitions while supporting long-horizon rollouts.

**Audience:**

Yes

**Audience Explanation:**

This paper can be of interest for researchers working in the fields of system identification, model-based control and meta learning.

**Claims And Evidence:**

No

**Claims Explanation:**

The experiments do not cleanly isolate which component of the framework is responsible for the empirical improvements, and the theoretical stability guarantees are not fully supported by the implemented training procedure.

**The contribution of the ICI structure is not isolated.**
The paper clearly states that "The ICI structure accounts for the feedback mechanism and mitigates feedback-induced bias, while meta-learning enables the operator to be adapted from limited target data rather than trained independently from scratch". The comparisons are entirely against ICI-based baselines (e.g., ICI-meta vs. ICI-scratch vs. ICI-full-data) and the experiments do not isolate the contribution of ICI from the contribution of meta-learning. A non-ICI meta-learning baseline would be needed to highlight the unique value of this combination, proving that integrating the ICI structure yields performance gains beyond what meta-learning provides on its own. Moreover, the theoretical justification relies on a high-SNR linearization ($\\hat{y} \\approx S(r) + S_L(K_L(v)) + v$), but there is no quantification of the residual bias in the few-shot regime.

**Soft constraints weaken the formal stability guarantees.**
The Schur-Koopman (SK) theorem requires strict assumptions about contraction of the encoder'decoder composition, while the training objective only encourages these conditions indirectly through a soft average $\\ell_2$ reconstruction loss. The authors must be more careful about distinguishing what is mathematically *guaranteed* by the theorem versus what is merely *encouraged* by the training.

**Secondary Empirical Issues:**
The Scratch-SK comparison is not fully diagnostic and is empirically confounded. Fast Meta-ICI uses both a meta-learned initialization and learned inner-loop step sizes via Meta-SGD, while Scratch-SK uses neither. As a result, if Scratch-SK diverges, it does not tell us whether the missing ingredient is the meta-initialization, the learned step sizes, or both. This weakens any causal interpretation of the comparison.
Additionally, both experiments evaluate a single target system, so "adapts to a new target" is single-target evidence. For Fast Meta-ICI specifically, the reported variance only covers five adaptation-data seeds on a single meta-checkpoint, excluding meta-training randomness and overstating the model's robustness to genuinely new targets.

**Requested Changes:**

* **State the contribution sharply and delineate it from ICI.** The boundary between inherited material and new contributions is not completely clear: the authors should add an explicit statement separating them.

* **Isolate the Contribution Empirically or Rescope Claims.** The authors should add a non-ICI meta-learning baseline (e.g., MAML on a plain neural state-space model) to isolate the value of ICI in the identification procedure.

* **Acknowledge the limitations of $\\mathcal{L}_p$-Stability Guarantees.** The authors must explicitly acknowledge that the $\\mathcal{L}_p$ stability of the Schur-Koopman operator is only encouraged empirically, not strictly guaranteed. While the spectral-radius condition of the latent transition matrix is enforced by the Schur parameterization, the contraction condition for the nonlinear encoder-decoder pair is only softly promoted via the Euclidean reconstruction loss. This does not constitute a hard guarantee that the residual will remain sufficiently small to preserve overall contraction.

* **Report variance over meta-training seeds:** The reported $\\pm$ values span five adaptation-data seeds with a single meta-trained checkpoint, so meta-training randomness is excluded and total uncertainty is understated. Add meta-training seeds, or state this limitation.

* **Symbol overloading:** $M$ (number of source tasks, Sec. 3) vs. $M$ (bound on $\\|g_\\theta\\|$, Thm. 1); $T$ (horizon) vs. $T_\\theta$ (Schur matrix).

---

> ### Author Response · Authors · 2026-07-03
> **Responses to the comments from Reviewer 5pr5 (Part 1/2)**
>
> We thank the reviewer for the rigorous evaluation of our work. We have revised the manuscript to clarify the contribution boundary, distinguish hard guarantees from empirical regularization, and clarify the scope of the empirical evidence. We address detailed concerns point by point.
>
> > 1. Non-ICI Meta learning baseline
>
> The purpose of our experiments is to evaluate whether meta-learning can reduce the target-data requirement of an ICI-based closed-loop identification procedure.
>
> In our setting, ICI plays the role of the closed-loop identification mechanism. It provides a way to construct an open-loop model from closed-loop data through the ICI interconnection.
> Meta-learning has a different role.
> It does not by itself define such an identification mechanism.
> Instead, it learns an initialization and adaptation rule for a given task family.
> Therefore, removing ICI while keeping only meta-learning would no longer give a well-defined baseline for the same closed-loop open-loop reconstruction problem, unless another closed-loop identification framework is introduced.
> A plain MAML neural state-space model would not use the ICI interconnection to reconstruct the open-loop plant from feedback-controlled data.
>
> This is why our empirical design compares Full-Data ICI, Scratch ICI, and Meta-ICI.
> The Full-Data ICI reference shows what the same identification structure can achieve with sufficient target data.
> Scratch ICI shows that the same structure is difficult to adapt from scarce target data alone.
> Meta-ICI then tests whether meta-learning can make this ICI-based identification structure effective in the few-shot regime.
>
> We have revised Section 4.1 to make this scope explicit.
> Our claim is that meta-learning improves the few-shot usability of ICI-compatible closed-loop identification.
>
> > 2. High-SNR linearization
>
> In our experiments, the noise level is treated as a hyperparameter setting of the ICI-based identification procedure.
> The ICI construction requires sufficiently informative perturbations around the closed-loop trajectory, while the effective range is system-dependent because the local linearization residual depends on the plant nonlinearity, the controller, and the operating region.
> Thus, for each experiment, we selected this hyperparameter by checking that the full-data ICI reference can converge and reconstruct the open-loop behavior under the chosen setting.
>
> Within each experiment, Meta-ICI and the corresponding full-data and scratch baselines use the same noise configuration.
> Therefore, the performance differences are evaluated under the same ICI-compatible closed-loop identification setting.
>
> Our goal is not to characterize the full robustness range of ICI with respect to SNR.
> Instead, we use an ICI-compatible regime and study whether meta-learning can reduce the amount of target-system data required in that regime.
>
> > 3. Soft constraints
>
> We agree that this distinction should be made explicit.
> The SK parameterization and the reconstruction loss play different roles in our implementation.
> The SK parameterization imposes a hard constraint on the latent linear dynamics and enforces its stability by construction.
> The reconstruction loss does not provide an exact mathematical constraint on the encoder-decoder consistency.
> It is a soft penalty used during training to encourage this consistency.
> Accordingly, our theoretical statement relies on the stability enforced by the SK parameterization, while the encoder-decoder consistency in the trained neural model should be understood as an approximation encouraged by the loss.
> In section 4.2.1, we have revised the sentence to make this distinction clear.
>
> > 4. Scratch-SK comparison
>
> Scratch-SK is not intended as an ablation study of the meta-learning algorithm.
> Its role is to test whether the same SK-parameterized model can be trained directly from scarce target-system data.
>
> In this comparison, Scratch-SK and Fast Meta-ICI use the same SK model structure and the same amount of target-system adaptation data.
> The key difference is that Fast Meta-ICI uses information learned from the task family, including the initialization and the adaptation rule, while Scratch-SK is trained only from the target data.
> This distinction is what the comparison is designed to evaluate.
>
> Meta-SGD is part of the meta-learning procedure.
> The learned step sizes are obtained during meta-training together with the initialization.
> Therefore, applying them to Scratch-SK without meta-training would not define a scratch baseline anymore.
>
> The comparison with the full-data SK reference further clarifies the issue.
> The same SK model can perform well when sufficient target data are available, while direct training from scarce target data is ineffective.
> This supports our interpretation that the main difficulty of Scratch-SK comes from data scarcity in the target system, and that Fast Meta-ICI improves adaptation by transferring useful information from related systems.

---

> ### Author Response · Authors · 2026-07-03
> **Responses to the comments from Reviewer 5pr5 (Part 2/2)**
>
> > 5. Single target system problem
>
> Each experiment evaluates one specified target system together with its excitation, perturbation, and noise setting.
> This protocol follows common practice in meta-learning-based system identification, where a specified target system is used to compare different adaptation strategies under a fixed experimental design.
> For example, [1],[2] evaluate meta-learned neural state-space models on specified target systems and compare different learning strategies under the same target-system protocol.
>
> This design is important because the excitation and noise settings are part of the identification benchmark.
> Results from different target systems cannot be directly pooled as independent random seeds.
> If each target uses its own excitation and noise configuration, the resulting errors also reflect differences in the experiment design.
> If the same excitation is forced on all targets, it may correspond to very different excitation levels for different systems.
>
> Therefore, for each experiment, we fix the target system and its excitation and noise setting, and compare all methods under exactly the same protocol.
> This gives a controlled comparison of whether meta-learning can reduce the amount of target-system data required by the ICI or SK identification model.
>
> ---
>
> [1] Chakrabarty, Ankush, Gordon Wichern, and Christopher R. Laughman. "Meta-learning of neural state-space models using data from similar systems." IFAC-PapersOnLine 56.2 (2023): 1490-1495.
>
> [2] Chakrabarty, Ankush, et al. "Meta-learning for physically-constrained neural system identification." Neurocomputing (2025): 130945.
>
> > 6. Contribution boundary between ICI and Meta-ICI
>
> The boundary between inherited ICI components and our contribution is now stated more explicitly in the revised manuscript.
>
> The ICI interconnection and the associated closed-loop identification principle are inherited from the original ICI framework.
> We do not claim these components as new contributions.
> Our contribution is to use ICI as the closed-loop identification backbone and introduce a meta-learning framework on top of it, so that the ICI-based model can be adapted to a new target system from scarce target-system data.
> In the Fast Meta-ICI variant, we further combine this idea with the SK parameterization and a meta-learned adaptation rule to enable efficient few-shot adaptation from fragmented target transitions.
>
> In the revised Section 4.1, we have clarified the source and role of the ICI formulation and explained how it serves as the basis for the proposed Meta-ICI framework.
>
> > 7. Variance over meta-training seeds
>
> The reported variance in Exp. 2 is over target adaptation-data seeds for a fixed validation-selected meta-checkpoint and a fixed validation-selected adaptation step.
> It does not include meta-training randomness.
>
> In Exp. 2, the reported model is obtained through a serial meta-training and validation pipeline.
> We first construct a task family by sampling systems from the parameter space.
> For each task, trajectories are generated from different initial conditions.
> During meta-training, each update further involves random task sampling, random trajectory sampling, and random transition or window sampling for the inner and outer loops.
> After meta-training, the learned parameters are passed to a validation procedure, which selects both the final meta-checkpoint and the number of adaptation steps used for testing.
>
> Therefore, changing the random seed at any earlier stage does not only perturb one isolated factor.
> It changes the subsequent training trajectory and may also lead to a different selected checkpoint and a different validation-selected adaptation step.
> The resulting variance would then reflect the robustness of the entire serial meta-training and model-selection pipeline, rather than only the stability of few-shot target adaptation.
>
> In particular, different meta-training runs may select different optimal adaptation steps.
> For example, comparing a run that uses 5 adaptation steps with another run that uses 15 adaptation steps would mix the effect of the learned initialization, the learned adaptation rule, and the selected adaptation budget.
> Forcing all runs to use the same number of steps would also be unfair to checkpoints whose validation-selected adaptation step is different.
>
> Therefore, in the reported results, we fix the meta-checkpoint and the validation-selected adaptation step, and report variance over the target-system adaptation data.
> This statistic directly measures the stability of few-shot adaptation under the fixed meta-learned prior and fixed adaptation protocol, which is the focus of the experiment.
>
>
> > 8. Symbol overloading
>
> We have revised the notation to remove symbol overloading.

---

### Review · Reviewer_eCBT · 2026-06-21

**Summary Of Contributions:**

This work studies the combination of Internal Controller Identification (ICI) and meta learning to propose Meta-ICI.  It proposes stable dynamics learning using Koopman-inspired idea and parameterization through a Schur form.

**Audience:**

Yes

**Audience Explanation:**

The work is about sysid of controlled dynamics which in general is of interest of TMLR community.

**Claims And Evidence:**

No

**Claims Explanation:**

Logic around the method and its validation on simple experiments are fair.
However, the comparison to existing studies mentioned in the related work and its significance should be validated by clearer assumption/claim structures or math etc.

**Requested Changes:**

Some parts of the manuscript are hard to parse or unclear, which makes it hard to grasp the overall story and contributions (including but not limited to the relations of closed-loop/open-loop, POMDP/MDP etc.).

1. Please consider modifying Abstract to make it crystal clear what the problem is and the main (general/technical) contribution is.  E.g. relation between learning of closed-loop dynamics and that of open-loop dynamics is rather unclear.

2. It is not really clear what is the point of separating closed-loop and open-loop in this case; existing work on learning controlled dynamics should also account for open-loop dynamics in general (of course, there will be uncertainty around the unknown control inputs though).  Does the class of noise matter?  It says in page 4 that “unless additional assumptions, excitation, or identification structure are used”; but I think the clarification of this aspect is important to showcase the novelty of this work because this work anyway makes several assumptions as well.

3. Page 3, it says “While existing ..., their inherent reliance on contiguous data sequences to infer hidden states makes them inapplicable when only fragmented
data points are available for few-shot target adaptation.  For the fully observable setting, we overcome…”; so those two work consider different setups anyway?  (Partially observable vs Fully observable cases?)

4. Assuming that the controller K is known seems a bit strong; do you have any good reasoning around this?   Also, K* is known; it would be better to summarize all the assumptions in the assumption environments.

5. The explanations about eq. 5 and 6 are not sufficient; it would be better to put figures etc. to clarify them because it seems the core starting point of the discussions.  Also, while this manuscript mentions in related work about the relation to Boroujeni et al., it would be better to clarify more about what the core additions of this work are.

6. Related to the above; I still cannot see what would be the core difference/advantage of this work in relation to existing work; does this method have sample complexity, robustness analyses etc.?

7. How restrictive of the form (8) compared to the general dynamics f?  This is an important issue when considering Koopman operators for controlled dynamics.  This is again a strong assumption isn’t it?  This also leads to the validity of comparison conducted in the experiment section.

Minor
1. page 5: …, satisfies → satisfying ?


In general, I still cannot clearly see how the structural assumptions etc. are different/same compared to existing work; which makes it hard to verify what would be novel about the work.

---

> ### Author Response · Authors · 2026-07-03
> **Responses to the comments from Reviewer eCBT (Part 1/2)**
>
> We thank the reviewer for the detailed and constructive comments.
> We have revised the manuscript to clarify the problem statement, the closed-loop/open-loop distinction, the role of stable neural operators, the assumptions, and the scope of the contribution.
> We address the comments below.
>
> > 1. Abstract
>
> We have revised the abstract to more clearly state the problem, the closed-loop data-generation challenge, and the main contributions of Meta-ICI and Fast Meta-ICI.
>
> > 2. The point of separating closed-loop and open-loop
>
> The purpose of separating closed-loop and open-loop is to distinguish the data-generation mechanism from the identification target.
> The target of system identification is still the plant dynamics $G$.
>
> In an open-loop experiment, there is no feedback controller in the data-generating loop, and the input is externally applied to the plant.
> Therefore, learning from the measured input-output data can be interpreted directly as learning the plant map.
>
> In a closed-loop experiment, the plant input is generated through the feedback controller from the measured output.
> We have added a new figure in Section 4.1 to illustrate this process.
> As shown in the left panel of Fig. 1, the observed trajectory is produced by the interconnection among the plant, the controller, and the noise path.
>
> If such data are treated as ordinary supervised input-output data, the fitted model can absorb feedback-induced correlations and noise effects, instead of representing the plant dynamics that can be used independently of the controller.
> This is why closed-loop identification is separated from standard open-loop identification.
> Since most practical controlled systems operate under feedback, this distinction is central to the motivation of closed-loop system identification.
>
> > 3.  Different setup works
>
> The two cited works are related to our paper only at the level of stable neural operator design.
> They are not direct predecessors of our main problem, which is few-shot closed-loop system identification with meta-learning.
> Thus, the distinction between our work and these stable-dynamics-learning works should not be understood as a difference between partially observable and fully observable problem settings.
>
> We cite these works because the ICI framework requires the learned internal operator to be stable.
> In the general partially observable case, a recurrent stable operator such as REN is suitable because continuous trajectories are available and the recurrent state can be propagated along the sequence.
>
> In the fragmented-adaptation setting of Fast Meta-ICI, the target data consist of scattered one-step transitions.
> Recurrent architectures are not directly suitable for this data format because they would treat unrelated samples as consecutive time steps.
> We therefore use the fully observable condition to avoid this issue and design a Markovian stable operator that can be adapted from independent transitions.
> This is the role of the Schur-Koopman operator in Fast Meta-ICI.
>
> > 4. Knwon controller and assumption env
>
> The known-controller assumption is standard in closed-loop identification.
> In many practical systems, the controller is designed, implemented, and tuned by the practitioner, while the plant dynamics usually remain unknown.
> For example, PID controller can be deployed and tuned without an accurate plant model, but their feedback laws and parameters are known.
> Therefore, assuming known $K_i$ and $K_\star$ does not imply knowing $G_i$ or $G_\star$.
>
> This assumption is also required by the ICI framework, where the internal operator is defined with respect to the known controller.
> In our setting, Meta-ICI learns an initialization for the ICI-induced internal operators, and the adapted operator is combined with the known target controller to construct the target open-loop model.
>
> Regarding assumption environments, we keep the assumptions in their relevant contexts because they serve different roles in the paper.
> The known-controller assumption belongs to the problem formulation, the ICI stability assumptions belong to the Meta-ICI construction, and the Schur-Koopman assumptions belong to the stability theorem.

---

> ### Author Response · Authors · 2026-07-03
> **Responses to the comments from Reviewer eCBT (Part 2/2)**
>
> > 5. Explanations about Eqs 5, 6
>
> We have added Fig. 1 in Section 4.1 to make the ICI construction easier to follow.
> We also expanded the text around Eq. (5) and Eq. (6) to clarify that Eq. (5) defines the interconnection between $S_i$ and $K_i$, while Eq. (6) defines the resulting open-loop input-output map $\hat G_i$.
>
> These revisions are intended to make the starting point of the Meta-ICI framework more self-contained.
>
> > 6. Core difference and advantage of this work
>
> The core difference of our work is that we address few-shot adaptation in closed-loop system identification, where the target plant must be recovered from scarce feedback-controlled data.
> Existing meta-learning methods for neural system identification mainly study adaptation from open-loop trajectory data.
> Existing ICI-style closed-loop identification methods provide a way to account for feedback-induced bias, but they are trained for each target system independently with enough data and do not address few-shot transfer across related closed-loop systems.
>
> Our contribution is to connect these two directions.
> Meta-ICI uses the ICI reparameterization to preserve the closed-loop identification structure, while using meta-learning to obtain an initialization that can be adapted from limited target data.
> This gives a practical advantage at the target-adaptation stage, since the target system does not need to be identified from scratch.
>
> Fast Meta-ICI further addresses a specific data limitation that is common in practice.
> When the target system is fully observable, adaptation can use scattered one-step transitions instead of continuous target trajectories.
> The Schur-Koopman operator is introduced to provide an ICI-compatible stable operator for this fragmented-adaptation setting.
>
> We do not claim formal sample-complexity or robustness bounds.
> The current paper supports the advantage empirically through few-shot adaptation results, fragmented-data adaptation, and long-horizon rollout evaluation under shifted input conditions.
> We have clarified the scope of the contribution accordingly.
>
> > 7. About eq 8
>
> This is a good question.
> Eq. (8) is more restrictive than a general nonlinear controlled dynamics model $f(x_t,r_t)$.
> The SK operator should be understood as an input-affine model with respect to the chosen input coordinate.
> Therefore, if the true dynamics depend on nonlinear raw-input terms such as $r_t^2$, $r_t^3$, or other nonlinear transformations of $r_t$, Eq. (8) may not represent such effects exactly when only the raw input $r_t$ is used.
>
> This restriction is a deliberate trade-off in the current Fast Meta-ICI instantiation.
> A fully general neural dynamics model could be more expressive, but it would not satisfy the stable-operator requirement needed by the ICI framework.
> In Eq. (8), the Schur-stable latent autonomous part and the bounded input map make the stability argument tractable, which is why Theorem 1 can establish an $L_p$ gain bound from the input sequence to the state sequence.
>
> Importantly, the input-affine form can be also defined with respect to the selected input coordinate.
> In control-oriented modeling, prior physical knowledge can often be used to transform or augment the input.
> For example, if the relevant actuation variable depends on $r_t^2$ or $r_t^3$, one can use an augmented input such as $[r_t,r_t^2,r_t^3]$, and then apply the same input-affine structure to that coordinate.
>
> Accordingly, we do not claim that Eq. (8) is a universal parameterization for arbitrary controlled dynamics.
> It is an ICI-compatible stable neural operator designed for the fully observable and fragmented-adaptation setting considered in Fast Meta-ICI.
> The experimental comparison should be interpreted within this scope.
> The Full-Data SK-ICI Reference and Fast Meta-ICI use the same SK model class, so the comparison evaluates whether meta-learning and fragmented adaptation can approach the performance of the same architecture trained with abundant target data, rather than whether the SK architecture is the most expressive model.
>
> > 8. Minor
>
> We have revised the wording accordingly.
>
> We hope that the above revisions and clarifications help address this general concern.

---

> > ### Comment · Reviewer_eCBT · 2026-07-03
> > **Thank you for the response**
> >
> > Thank you for the response; I will look into it.